# Membership Inference Attacks against Large Vision-Language Models

Zhan Li[*]  Yongtao Wu[*]  Yihang Chen[*][†]
Francesco Tonin  Elias Abad Rocamora  Volkan Cevher
LIONS, EPFL
[first name].[last name]@epfl.ch

## Abstract

Large vision-language models (VLLMs) exhibit promising capabilities for processing multi-modal tasks across various application scenarios. However, their emergence also raises significant data security concerns, given the potential inclusion of sensitive information, such as private photos and medical records, in their training datasets. Detecting inappropriately used data in VLLMs remains a critical and unresolved issue, mainly due to the lack of standardized datasets and suitable methodologies. In this study, we introduce the first membership inference attack (MIA) benchmark tailored for various VLLMs to facilitate training data detection. Then, we propose a novel MIA pipeline specifically designed for token-level image detection. Lastly, we present a new metric called `MaxRényi-K%`, which is based on the confidence of the model output and applies to both text and image data. We believe that our work can deepen the understanding and methodology of MIAs in the context of VLLMs. Our code and datasets are available at https://github.com/LIONS-EPFL/VL-MIA.

## 1 Introduction

The rise of large language models (LLMs) [9, 60, 45, 11] has inspired the exploration of large models across multi-modal domains, exemplified by advancements like GPT-4 [1] and Gemini [59]. These large vision-language models (VLLMs) have shown promising ability in various multi-modal tasks, such as image captioning [33], image question answering [13, 35], and image knowledge extraction [26]. However, the rapid advancement of VLLMs also causes user concerns about privacy and knowledge leakage. For instance, the image data used during commercial model training may contain private photographs or medical diagnostic records. This is concerning since early work has demonstrated that machine learning models can memorize and leak training data [3, 56, 63]. To mitigate such concerns, it is essential to consider the membership inference attack (MIA) [23, 53], where attackers seek to detect whether a particular data record is part of the training dataset [23, 53]. The study of MIAs plays an important role in preventing test data contamination and protecting data security, which is of great interest to both industry and academia [24, 19, 44].

When exploring MIAs in VLLMs, one main issue is the absence of a standardized dataset designed to develop and evaluate different MIA methods, which comes from the large size [16] and multi-modality of the training data, and the diverse VLLMs training pipelines [66, 35, 18]. Therefore, one of the main goals of this work is to build an MIA benchmark tailored for VLLMs.

Beyond the need for a valid benchmark, we lack efficient techniques to detect a single modality in VLLMs. The closest work to ours is [30], which performs MIAs on multi-modal CLIP [46] by detecting whether an image-text pair is in the training set. However, in practice, it is more common

---

[*]Equal contribution.

[†]Corresponding author. Email: yhangchen@cs.ucla.edu.

38th Conference on Neural Information Processing Systems (NeurIPS 2024).

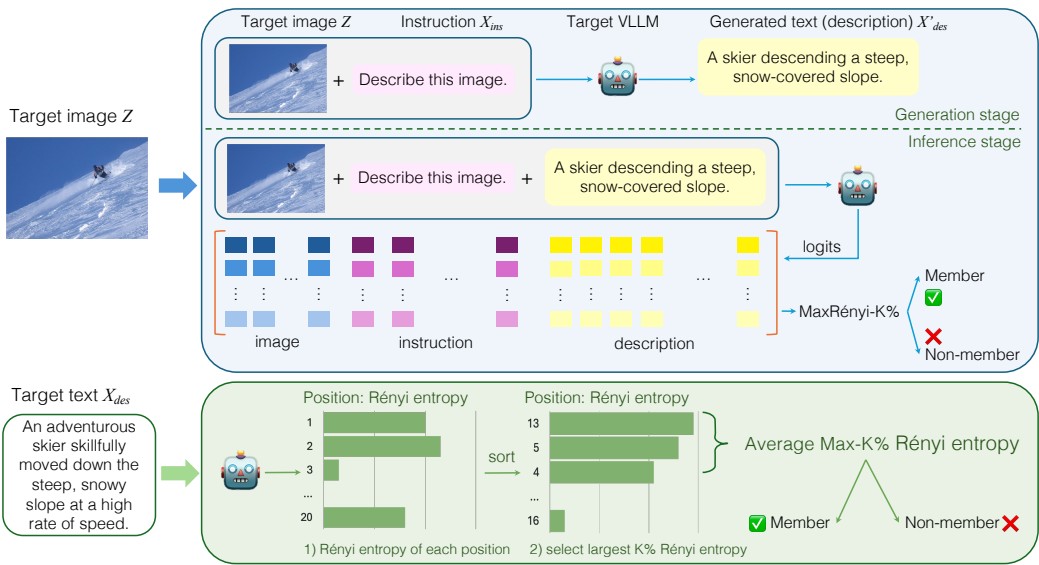

Figure 1: **MIAs against VLLMs**. **Top:** Our image detection pipeline: In the generation stage, we feed the image and instruction to the target model to obtain a description; then during the inference stage, we input the image, instruction, and generated description to the model, and extract the logits slices to calculate metrics. **Bottom:** `MaxRényi-K%` metric: we first get the Rényi entropy of each token position, then select the largest $k\%$ tokens and calculate the average Rényi entropy.

to detect a single modality, as we care whether an individual image or text is in the training set. Therefore, we aim to develop a pipeline to detect the single modality from a multi-modal model. Moreover, existing literature on language model MIAs, such as `Min-K%` [52] and `Perplexity` [62], mostly are *target-based* MIAs, which use the next token as the target to compute the prediction probability. However, we can only access the image embedding instead of the image token in VLLMs, and thus only *target-free* MIAs [48] can be directly applied.

Therefore, we first propose a cross-modal pipeline for individual image or description MIAs on VLLMs, which is distinguished from traditional MIAs that only use one modality [61, 62]. We feed the VLLMs with a customized image-instruction pair from the target image or description. We show that we can perform the MIA not only by the image slice but also by the instruction and description slices of the VLLM's output logits, see Figure 1. Such a cross-modal pipeline enables the usage of text MIA methods on image MIAs. We also introduce a target-free metric that adapts to both image and text MIAs and can be further modified to a target-based way.

Overall, the contributions and insights can be summarized as follows.

- We release the first benchmark tailored for the detection of training data in VLLMs, called **V**ision **L**anguage **MIA** (**VL-MIA**) (Section 4). By leveraging Flickr and GPT-4, we construct VL-MIA that contains two images MIA tasks and one text MIA task for various VLLMs, including MiniGPT-4 [66], LLaVA 1.5 [35] and LLaMA-Adapter V2 [18].

- We perform the first individual image or description MIAs on VLLMs in a cross-modal manner. Specifically, we demonstrate that we can perform image MIAs by computing statistics from the image or text slices of the VLLM's output logits (Figure 1 and Section 5.1).

- We propose a target-free MIA metric, `MaxRényi-K%`, and its modified target-based `ModRényi` (Section 5.2). We demonstrate their effectiveness on open-source VLLMs and closed-source GPT-4 (Section 6). We achieve an AUC of 0.815 on GPT-4 in image MIAs.

## 2   Related work

**Membership Inference Attack (MIA)** aims to classify whether a data sample has been used in training a machine learning model [53]. Keeping training data confidential is a desired property for

machine learning models, since training data may contain private information about an individual [62, 24]. Popular MIA methods can be divided into metric-based and shadow model-based MIAs [24]. Metric-based MIAs [62, 48, 57, 52] determine whether a data sample has been used for training by comparing metrics computed from the output of the target model with a threshold. Shadow model-based MIAs need shadow training to mimic the behavior of the target model [53, 48], which is computationally infeasible for LLMs. Therefore, we focus on the metric-based methods in this work.

MIAs have been extensively researched in various machine learning models, including classification models [38, 58, 10], generative models [20, 22, 7], and embedding models [54, 40]. With the emergence of LLMs, there are also a lot of work exploring MIAs in LLMs [42, 17, 52, 41]. Nevertheless, MIAs for multi-modal models have not been fully explored. [30, 25] perform MIAs using the similarity between the image and the ground truth text, which detects the image-text pair instead of a single image or text sequence. However, detecting an individual image or text is more practical in real-world scenarios and poses additional challenges. To the best of our knowledge, we are the first to perform the individual image or text MIA on VLLMs.

In this paper, we also consider a more difficult task, to detect the pre-training data from the fine-tuned models, that is, to detect the base LLM pertaining data from VLLMs. First, compared with detecting fine-tuning data [55, 51], pretraining data come from a much larger dataset and are used only once, reducing the potential probability for a successful MIA [27, 32]. In addition, compared to the detection of pretraining data from the pre-trained models [53, 52], catastrophic forgetting [29, 28, 16] in the fine-tuning stage also makes it harder to detect the pre-training data from downstream models. To the best of our knowledge, we are the first to perform pre-training data MIAs on fine-tuned models.

**Large Vision-Language Models (VLLMs)** incorporate visual preprocessors into LLMs [9, 60, 45] to manage tasks that require handling inputs from text and image modalities. A foundational approach in this area is represented by CLIP [46], which established techniques for aligning modalities between text and images. Further developments have integrated image encoders with LLMs to create enhanced VLLMs. These models are typically pre-trained on vast datasets of image-text pairs for feature alignment [33, 64, 39, 2], and are subsequently instruction-tuned for specific downstream tasks to refine the end ability. MiniGPT [66, 8], LLaVA [36, 35], and LLaMA Adapter [65, 18] series have demonstrated significant capabilities in understanding and inference in this area.

## 3 Problem setting

In this section, we introduce the main notation and problem settings for MIAs.

**Notation.** The token set is denoted by $\mathcal{V}$. A sequence with $L$ tokens is denoted by $X := (x_1, x_2, \ldots, x_L)$, where $x_i \in \mathcal{V}$ for $i \in [L]$. Let $X_1 \oplus X_2$ be the concatenation of sequence $X_1$ and $X_2$. An image token sequence is denoted by $Z$. In this work, we focus on the VLLM, parameterized by $\theta$, where the input is the image $Z$ followed by the instruction text $X_{\text{ins}}$, and the output is the description text $X_{\text{des}}$. We use $\mathcal{D}_{\text{des}}$ and $\mathcal{D}_{\text{image}}$ to represent the description training set and image training set, respectively. Detailed notations are summarized in Table 5 of the appendix.

**Attacker's goal.** In this work, the purpose of the attacker is to detect whether a given data point (image $Z$ or description $X_{\text{des}}$) belongs to the training set. We formulate this attack as a binary classification problem. Let $\mathbf{A}_{\text{image}}(Z; \theta) :\rightarrow \{0, 1\}$ and $\mathbf{A}_{\text{des}}(X; \theta) :\rightarrow \{0, 1\}$ be two binary classification algorithms for image and description respectively, which are implemented by comparing the metric $\texttt{Score}(Z \oplus X_{\text{ins}} \oplus X_{\text{des}}; \theta)$ with some threshold $\lambda$.

When detecting image $Z$, we feed the model with the target image with a fixed instruction prompt such as "Describe this image in detail", denoted as $X_{\text{ins}}$. The model then generates the description text $X'_{\text{des}}$. The algorithm $\mathbf{A}_{\text{image}}(Z; \theta)$ is defined by

$$\mathbf{A}_{\text{image}}(Z; \theta) = \begin{cases} 1 & (Z \in \mathcal{D}_{\text{image}}), \text{ if } \texttt{Score}(Z \oplus X_{\text{ins}} \oplus X'_{\text{des}}; \theta) < \lambda, \\ 0 & (Z \notin \mathcal{D}_{\text{image}}), \text{ if } \texttt{Score}(Z \oplus X_{\text{ins}} \oplus X'_{\text{des}}; \theta) \geq \lambda. \end{cases} \quad (1)$$

When detecting a description sequence $X_{\text{des}}$, we feed the model with an all-black image, denoted as $Z_{\text{ept}}$, as the visual input, followed by an empty instruction $X_{\text{ept}}$. The algorithm $\mathbf{A}_{\text{des}}(X_{\text{des}}; \theta)$ is defined by

$$\mathbf{A}_{\text{des}}(X_{\text{des}}; \theta) = \begin{cases} 1 & (X_{\text{des}} \in \mathcal{D}_{\text{des}}), \text{ if } \texttt{Score}(Z_{\text{ept}} \oplus X_{\text{ept}} \oplus X_{\text{des}}; \theta) < \lambda, \\ 0 & (X_{\text{des}} \notin \mathcal{D}_{\text{des}}), \text{ if } \texttt{Score}(Z_{\text{ept}} \oplus X_{\text{ept}} \oplus X_{\text{des}}; \theta) \geq \lambda. \end{cases} \quad (2)$$

**Attacker's knowledge.** We assume a grey-box setting on the target model, where the attacker can query the model by a custom prompt (including an image and text) and have access to the tokenizer, the output logits, and the generated text. The attacker is unaware of the training algorithm and parameters of the target model.

# 4 Dataset construction

We construct a general dataset: **V**ision **L**anguage **MIA** (**VL-MIA**), based on the training data used for popular VLLMs, which, to our knowledge, is the first MIA dataset designed specifically for VLLMs. We present a takeaway overview of VL-MIA in Table 1. We also provide some examples in VL-MIA, see Table 16 in the appendix. The prompts we use for generation can be found in Table 6.

Table 1: **Overview of VL-MIA dataset**: VL-MIA covers image and text modalities and can be applied for dominant open-sourced VLLMs.

| Dataset | Modality | Member data | Non-member data | Application |
|---------|----------|-------------|-----------------|-------------|
| VL-MIA/DALL-E | image | LAION_CCS | DALL-E-generated images | LLaVA 1.5 MiniGPT-4 LLaMA_adapter v2 |
| VL-MIA/Flickr | image | MS COCO (from Flickr) | Latest images on Flickr | LLaVA 1.5 MiniGPT-4 LLaMA_adapter v2 |
| VL-MIA/Text | text | LLaVA v1.5 instruction-tuning text | GPT-generated answers | LLaVA 1.5 LLaMA_adapter v2 |
| | | MiniGPT-4 instruction-tuning text | GPT-generated answers | MiniGPT-4 |

**Target models.** We perform MIAs open-source VLLMs, including MiniGPT-4 [66], LLaVA-1.5 [35], and LLaMA-Adapter V2.1 [18]. The checkpoints and training datasets of these models are provided and public. The training pipeline of a VLLM encompasses several stages. Initially, an LLM undergoes pre-training using extensive text data such as LLaMA [18]. Meanwhile, a vision preprocessor, e.g., CLIP [46], is pre-trained on a large number of image-text pairs. Subsequently, a VLLM is constructed based on the LLM and the vision preprocessor and is pre-trained using image-text pairs. The final stage involves instruction-tuning the VLLM, which can be performed using either image-text pairs or image-based question-answer data. In the instruction tuning stage of a VLLM, every data entry contains an image, a question, and the corresponding answer to the image. We use the answer text as member data and GPT-4 generated answers under the same question and same image as non-member data. Specifically, for LLaVA 1.5 and LLaMA-Adapter v2, we use the answers in LLaVA 1.5's instruction tuning as member data.

**VL-MIA/DALL-E.** MiniGPT-4, LLaVA 1.5, and LLaMA-Adapter V2 use images from LAION [49], Conceptual Captions 3M [6], Conceptual 12M [6] and SBU captions [43] datasets (collectively referred to as LAION-CCS) as pre-training data. BLIP [33] provides a synthetic dataset with image-caption pairs for LAION-CCS used in MiniGPT-4 and LLaVA 1.5. We first detect the intersection of the training images used in these three VLLMs. From this intersection, we randomly select a subset to serve as the member data for our benchmark. For the non-member data, we use the corresponding BLIP captions of the member images as prompts to generate images with DALL-E 2[3]. This process yields one-to-one corresponding pairs of the generated images (non-member) and the original member images. Consequently, our dataset comprises an equal number of member images from the LAION-CCS dataset and non-member images generated by DALL·E, allowing us to evaluate MIA performance comprehensively. We have 592 images in VL-MIA/DALL-E in total.

**VL-MIA/Flickr.** MS COCO [34] co-occurs as a widely used dataset in the training data of the target models, so we use the images in this dataset as member data. Given the fact that such member data are collected from Flickr[4], we filter Flickr photos by the year of upload and obtain new photos from January 1, 2024, as non-member data, which are later than the release of the target models. We additionally prepare a set of corrupted versions, where the member images are deliberately corrupted

---

[3]https://platform.openai.com/docs/models
[4]https://www.flickr.com/

to simulate real-world settings. More results of the corrupted versions are discussed in Section 6.5. This dataset contains 600 images.

**VL-MIA/Text.** We prepare text MIA datasets for the VLLMs instruction-tuning stage. LLaVA 1.5 and LLaMA Adapter v2.1 both use the LLaVA-Instruct-150K [35] in instruction-tuning, which consists of multi-round QA conversations. We first select entries with descriptive answers of 64 words. Next, we feed the corresponding questions and images into GPT-4[3], and ask GPT-4 to generate responses of the same length, treating these generated responses as non-member text data. In addition, since MiniGPT-4 employs long descriptions of images for instruction-tuning, typically beginning with "the image", we prompt GPT to generate descriptions based on the MS COCO dataset, starting with "this image" to ensure similar data distributions. We also prepare different versions of the datasets by truncating the text into different word lengths such as 16, 32, and 64. Each text dataset contains 600 samples.

**VL-MIA/Synthetic** We synthesize two new MIA datasets: **VL-MIA/Geometry** and **VL-MIA/Password**. The image in the VL-MIA/Geometry consists of a random 4x4 arrangement of geometrical shapes, and the image in the VL-MIA/Password consists of a random 6x6 arrangement of characters and digits from MNIST [15] and EMINST [12]. The associated text for each image represents its content (e.g., specific characters, colors, or shapes), ordered from left to right and top to bottom. Half of the dataset can be selected as the member set for VLLM fine-tuning, with the remainder as the non-member set. This partition ensures that members and non-members are independently and identically distributed, avoiding the latent distribution shift between the members and non-members in current MIA datasets [14]. Our synthetic datasets are ready to use, and can be applied to evaluate any VLLM MIA methods through straightforward fine-tuning. We provide some examples of this dataset in Figure 5 in the Appendix C.

## 5 Method

### 5.1 A cross-modal pipeline to detect image

VLLMs such as LLaVA and MiniGPT project the vision encoder's embedding of the image into the feature space of LLM. However, a major challenge for image MIA is that we do not have the ground-truth image tokens. Only the embeddings of images are available, which prevents directly transferring many target-based MIA from languages to images. To this end, we propose a token-level image MIA which calculates metrics based on the output logit of each token position.

This pipeline consists of two stages, as demonstrated in Figure 1. In *generation stage*, we provide the model with an image followed by an instruction to generate a textual sequence. Subsequently, in *inference stage*, we feed the model with the concatenation of the same image, instruction, and generated description text. During the attack, we correspondingly slice the output logits into image, instruction, and description segments, which we use to compute various metrics for MIAs. Our pipeline considers the information from the image, the instructions and the descriptions following the image. In practice, even if there is no access to the logits of the image feature and instruction slice, we can still detect the member image solely from the model generation. We visually describe a prompt example with different slice notations presented in Appendix A.2.

Our pipeline operates on the principle that VLLMs' responses always follow the instruction prompt [60], where the images usually precede the instructions and then always precede the descriptions. For causal language models used in VLLMs that predict the probability of the next token based on the past history [45], the logits at text tokens in the sequence inherently incorporate information from the preceding image.

### 5.2 MaxRényi MIA

We propose our `MaxRényi-K%`, utilizing the Rényi entropy of the next-token probability distribution on each image or text token. The intuition behind this method is that if the model has seen this data before, the model will be more confident in the next token and thus have smaller Rényi entropy.

Given a probability distribution $p$, the Rényi entropy [47] of order $\alpha$, is defined as $H_\alpha(p) = \frac{1}{1-\alpha} \log \left( \sum_j (p_j)^\alpha \right), 0 < \alpha < \infty, \alpha \neq 1$. $H_\alpha(p)$ is further defined at $\alpha = 1, \infty$, as $H_\alpha(p) = \lim_{\gamma \to \alpha} H_\gamma(p)$ by,

- $H_1(p) = -\sum_j p_j \log p_j,$    - $H_\infty(p) = -\log \max p_j.$

To be more specific, given a token sequence $X := (x_1, x_2, \ldots, x_L)$, let $p^{(i)}(\cdot) = \mathbb{P}(\cdot | x_1, \cdots, x_i)$ be the probability of next-token distribution at the $i$-th token. Let Max-K%$(X)$ be the top K% from the sequence $X$ with the largest Rényi entropies, the `MaxRényi-K%` score of $X$ equals

$$\mathtt{MaxR\acute{e}nyi\text{-}K\%}(X) = \frac{1}{|\text{Max-K\%}(X)|} \sum_{i \in \text{Max-K\%}(X)} H_\alpha(p^{(i)}).$$

When $K = 0$, we define the `MaxRényi-K%` score to be $\max_{i \in [L-1]} H_\alpha(p^{(i)})$. When $K = 100$, the `MaxRényi-K%` score is the averaged Rényi entropy of the sequence $X$.

In our experiments, we vary $\alpha = \frac{1}{2}, 1, 2$, and $+\infty$; $K = 0, 10, 100$. As $\alpha$ increases, the top percentile of distribution $p$ will have more influence on $H_\alpha(p)$. When $\alpha = 1$, $H_1(p)$ equals the Shannon entropy [50], and our method at $K = 100$ is equivalent to the `Entropy` [48]. When $\alpha = \infty$, we consider the most likely next token probability [31]. In contrast, `Min-K%` [52] only deals with the target next token probability. When the sequence is generated by the target model deterministically, i.e., when the model always generates the most likely next token, our `MaxRényi-K%` at $\alpha = \infty$ is equivalent to the `Min-K%`.

We also extend our `MaxRényi-K%` to the target-based scenarios, denoted by `ModRényi`. We first consider linearized Rényi entropy, $\overline{H}_\alpha(p) = \frac{1}{1-\alpha}\left(\sum_j (p_j)^\alpha - 1\right), 0 < \alpha < \infty, \alpha \neq 1$. $\overline{H}_\alpha(p)$ is also further defined at $\alpha = 1$, as $\overline{H}_1(p) = \lim_{\alpha \to 1} \overline{H}_\alpha(p) = H_1(p)$. Assuming the next token ID is $y$, recall that a small entropy value or a large $p_y$ value indicates membership, we want our modified entropy to be monotonically decreasing on $p_y$ and monotonically increasing on $p_j, j \neq y$. Therefore, we propose the modified Rényi entropy on a given next token ID $y$, denoted by $\overline{H}_\alpha(p, y)$:

$$\overline{H}_\alpha(p, y) = -\frac{1}{|\alpha - 1|}\left((1-p_y)p_y^{|\alpha-1|} - (1-p_y) + \sum_{j \neq y} p_j(1-p_j)^{|\alpha-1|} - p_j\right).$$

Let $\alpha \to 1$, we have $\overline{H}_1(p, y) = \lim_{\alpha \to 1} \overline{H}_\alpha(p, y) = -\sum_{j \neq y} p_j \log(1-p_j) - (1-p_y)\log p_y$, which is equivalent to the `Modified Entropy` [58]. In addition, our more general method does not encounter numerical instability in `Modified Entropy` as $p_j \to 0, 1$ at $\alpha \neq 1$. For simplicity, we let the `ModRényi` score be the averaged modified Rényi entropy of the sequence.

## 6 Experiments

In this section, we conduct MIAs across three target models using various baselines, `MaxRényi-K%`, and `ModRényi`. Experiment setup is provided in Section 6.1. The results on text MIAs and image MIAs are present in Section 6.2 and Section 6.3, respectively. In Section 6.4, we show that the proposed MIA pipeline can also be used in GPT-4. Ablation studies are present in Section 6.5. The versions and base models of VLLMs we use are listed in Table 7 of the appendices.

### 6.1 Experimental setup

**Evaluation metric.** We evaluate different MIA methods by their AUC scores. AUC score is the area under the receiver operating characteristic (ROC) curve, which measures the overall performance of a classification model in all classification thresholds $\lambda$. The higher the AUC score, the more effective the attack is. In addition to the average-case metric AUC, we also include the worst-case metric, the True Positive Rate at 5% False Positive Rate (TPR@5%FPR) in Appendix D suggested by [5].

**Baselines.** We take existing metric-based MIA methods as baselines and conduct experiments on our benchmark. We use the MIA method from [37], which compares the feature vectors produced by the original image with the augmented image. We use KL-divergence to compare the logit distributions and term it `Aug-KL` in this paper. We also use `Loss` attack [62], which is `perplexity` in the case of language models. Furthermore, we consider `ppl/zlib` and `ppl/lowercase` [4], which compare the target perplexity to zlib compression entropy and the perplexity of lowercase texts respectively. [52] proposes `Min-K%` method, which calculates the smallest $K\%$ probabilities corresponding to the

ground truth token. `Min-K%` is currently a state-of-the-art method to detect pre-training data of LLMs. For both `Min-K%` and `MaxRényi-K%`, we vary $K = 0, 10, 100$. In addition, we consider $K = 20$ for `Min-K%` as suggested in [52]. We further include the `max_prob_gap` metric that can represent the extreme confidence in certain tokens by the model. That is, we subtract the second largest probability from the maximum probability in each token position and calculate the mean as metric.

## 6.2 Image MIA

We first conduct MIAs on images using VL-MIA/Flickr and VL-MIA/DALL-E in three VLLMs. For the image slice, it is not possible to perform target-based MIAs, because of the absence of ground-truth token IDs for the image. However, our MIA pipeline presented in Figure 1 can still handle target-based metrics by accessing the instruction slice and description slice.

As demonstrated in Table 2, `MaxRényi-K%` surpasses other baselines in most scenarios. An $\alpha$ value of 0.5 yields the best performance in both VL-MIA/Flickr and VL-MIA/DALL-E. As $\alpha$ increases, performance becomes erratic and generally deteriorates, though it remains superior to all target-based metrics. Overall, target-free metrics outperform target-based metrics for image MIAs. Another interesting observation is that instruction slices result in unstable AUC values, sometimes falling below 0.5 in target-based MIAs. This can be partially explained by the fact the model is more familiar with the member data. As a result, after encountering the first word "Describe", the model is more inclined to generate the description directly than generating the following instruction of $X_{\text{ins}}$, i.e., "this image in detail". This is an interesting phenomenon that we leave to future research.

The performance of the image MIA model is influenced by its training pipelines. Recall that MiniGPT-4 only updates the parameters of the image projection layer in image training, and LLaMA Adapter v2 applies parameter-efficient fine-tuning approaches. In contrast, LLaVA 1.5 training updates both the parameters of the projection layer and the LLM. The inferior performance of MIAs on MiniGPT-4 and LLaMA Adapter compared to LLaVA 1.5 is therefore consistent with [52] that more parameters' updates make it easier to memorize training data.

We find that VL-MIA/DALL-E is a more challenging dataset than VL-MIA/Flickr, reflected in the AUC being closer to 0.5. In VL-MIA/DALL-E, each non-member image is generated based on the description of a member image. Therefore, member data have a one-to-one correspondence with non-member data and depict a similar topic, which makes it harder to discern.

## 6.3 Text MIA

Text member data might be used in different stages of VLLM training, including the base LLM model pre-training and the later VLLM instruction-tuning. We hypothesize that after the last usage of the member data in its training, the more the model changes, the better the target-free MIA methods compared to target-based ones, and vice-versa. The heuristic is that if the model's parameters have changed a lot, target-free MIA methods, which use the whole distribution to compute statistics, are more robust than target-based methods, which rely on the probability at the next token ID. On the other hand, if the member data are seen in recent fine-tuning, the next token will convey more causal relations in the sequence remembered by the model, and thus target-based ones are better.

Table 3: **Text MIA.** AUC results on LLaVA.

| Metric | | VLLM Tuning | | LLM Pre-Training | | | |
|---|---|---|---|---|---|---|---|
| | | 32 | 64 | 32 | 64 | 128 | 256 |
| Perplexity* | | 0.779 | 0.988 | 0.542 | 0.505 | 0.553 | 0.582 |
| Perplexity/zlib* | | 0.609 | 0.986 | 0.56 | 0.537 | 0.581 | 0.603 |
| Perplexity/lowercase* | | **0.962** | 0.977 | 0.493 | 0.518 | 0.503 | 0.583 |
| Min_0% Prob* | | 0.522 | 0.522 | 0.455 | 0.451 | 0.425 | 0.448 |
| Min_10% Prob* | | 0.461 | 0.883 | 0.468 | 0.487 | 0.526 | 0.534 |
| Min_20% Prob* | | 0.603 | 0.980 | 0.505 | 0.498 | 0.549 | 0.562 |
| Max_Prob_Gap | | 0.461 | 0.545 | 0.574 | 0.544 | 0.565 | 0.629 |
| | $\alpha = 0.5$ | 0.809 | 0.979 | 0.557 | 0.500 | 0.536 | 0.567 |
| ModRényi* | $\alpha = 1$ | 0.808 | **0.993** | 0.544 | 0.503 | 0.546 | 0.567 |
| | $\alpha = 2$ | 0.779 | 0.963 | 0.559 | 0.497 | 0.529 | 0.560 |
| | Max_0% | 0.506 | 0.514 | 0.541 | 0.515 | 0.489 | 0.571 |
| Rényi ($\alpha = 0.5$) | Max_10% | 0.458 | 0.776 | 0.518 | 0.525 | 0.606 | 0.65 |
| | Max_100% | 0.564 | 0.835 | 0.555 | 0.531 | 0.6 | 0.631 |
| | Max_0% | 0.552 | 0.579 | 0.566 | 0.571 | 0.603 | 0.668 |
| Rényi ($\alpha = 1$) | Max_10% | 0.566 | 0.809 | 0.553 | 0.541 | 0.623 | 0.65 |
| | Max_100% | 0.554 | 0.750 | 0.544 | 0.523 | 0.588 | 0.621 |
| | Max_0% | 0.589 | 0.625 | 0.594 | 0.606 | 0.659 | 0.657 |
| Rényi ($\alpha = 2$) | Max_10% | 0.607 | 0.787 | 0.583 | 0.556 | 0.629 | 0.663 |
| | Max_100% | 0.553 | 0.709 | 0.592 | 0.576 | 0.568 | 0.649 |
| | Max_0% | 0.600 | 0.638 | **0.607** | **0.615** | **0.688** | **0.669** |
| Rényi ($\alpha = \infty$) | Max_10% | 0.618 | 0.763 | 0.586 | 0.548 | 0.627 | 0.667 |
| | Max_100% | 0.557 | 0.694 | 0.546 | 0.527 | 0.584 | 0.634 |

Table 2: **Image MIA**. AUC results on VL-MIA under our pipeline. "img" indicates the logits slice corresponding to image embedding, "inst" indicates the instruction slice, "desp" the generated description slice, and "inst+desp" is the concatenation of the instruction slice and description slice. We use an asterisk $*$ in superscript to indicate the target-based metric. **Bold** indicates the best AUC within each column and underline indicates the runner-up.

| VL-MIA/Flickr | | | | | | | | | | | | |
|---|---|---|---|---|---|---|---|---|---|---|---|---|
| **Metric** | | **LLaVA** | | | | **MiniGPT-4** | | | | **LLaMA Adapter** | | |
| | | img | inst | desp | inst+desp | img | inst | desp | inst+desp | inst | desp | inst+desp |
| Perplexity* | | N/A | 0.378 | 0.667 | 0.559 | N/A | 0.414 | 0.649 | 0.497 | 0.380 | 0.661 | 0.425 |
| Min_0% Prob* | | N/A | 0.357 | 0.651 | 0.357 | N/A | 0.272 | 0.569 | 0.274 | 0.462 | 0.566 | 0.463 |
| Min_10% Prob* | | N/A | 0.357 | 0.669 | 0.390 | N/A | 0.272 | 0.603 | 0.265 | 0.437 | 0.591 | 0.438 |
| Min_20% Prob* | | N/A | 0.374 | 0.670 | 0.370 | N/A | 0.293 | 0.628 | 0.303 | 0.437 | 0.611 | 0.424 |
| Aug_KL | | 0.596 | 0.539 | 0.492 | 0.508 | 0.462 | 0.458 | 0.438 | 0.435 | 0.428 | 0.422 | 0.427 |
| Max_Prob_Gap | | 0.577 | 0.601 | 0.650 | 0.650 | 0.664 | 0.695 | 0.609 | 0.626 | 0.475 | 0.671 | 0.661 |
| ModRényi* | $\alpha = 0.5$ | N/A | 0.368 | 0.651 | 0.614 | N/A | 0.483 | 0.636 | 0.592 | 0.430 | 0.662 | 0.555 |
| | $\alpha = 1$ | N/A | 0.359 | 0.659 | 0.502 | N/A | 0.371 | 0.635 | 0.417 | 0.394 | 0.646 | 0.423 |
| | $\alpha = 2$ | N/A | 0.370 | 0.645 | 0.611 | N/A | 0.492 | 0.636 | 0.605 | 0.434 | 0.665 | 0.579 |
| Rényi ($\alpha = 0.5$) | Max_0% | 0.515 | 0.689 | 0.687 | 0.689 | 0.437 | 0.624 | 0.542 | 0.626 | 0.497 | 0.570 | 0.499 |
| | Max_10% | 0.557 | 0.689 | 0.691 | 0.719 | 0.493 | 0.624 | 0.592 | 0.707 | 0.432 | 0.573 | 0.622 |
| | Max_100% | **0.702** | **0.726** | **0.713** | 0.728 | **0.671** | **0.795** | **0.664** | 0.724 | **0.633** | 0.674 | 0.697 |
| Rényi ($\alpha = 1$) | Max_0% | 0.503 | 0.708 | 0.685 | 0.725 | 0.429 | 0.645 | 0.579 | 0.652 | 0.517 | 0.602 | 0.517 |
| | Max_10% | 0.623 | 0.708 | 0.698 | **0.743** | 0.489 | 0.645 | 0.627 | 0.710 | 0.456 | 0.610 | 0.565 |
| | Max_100% | 0.702 | 0.720 | 0.702 | 0.721 | 0.626 | 0.776 | 0.662 | **0.741** | 0.597 | **0.678** | 0.687 |
| Rényi ($\alpha = 2$) | Max_0% | 0.583 | 0.682 | 0.673 | 0.705 | 0.444 | 0.697 | 0.587 | 0.665 | 0.580 | 0.617 | 0.581 |
| | Max_10% | 0.621 | 0.682 | 0.685 | 0.725 | 0.482 | 0.697 | 0.621 | 0.734 | 0.499 | 0.615 | 0.584 |
| | Max_100% | 0.682 | 0.694 | 0.683 | 0.703 | 0.627 | 0.785 | 0.656 | 0.735 | 0.572 | 0.670 | 0.668 |
| Rényi ($\alpha = \infty$) | Max_0% | 0.588 | 0.646 | 0.651 | 0.674 | 0.462 | 0.693 | 0.569 | 0.657 | 0.604 | 0.566 | 0.603 |
| | Max_10% | 0.593 | 0.646 | 0.669 | 0.699 | 0.488 | 0.693 | 0.603 | 0.704 | 0.506 | 0.591 | 0.584 |
| | Max_100% | 0.669 | 0.673 | 0.667 | 0.687 | 0.632 | 0.769 | 0.649 | 0.725 | 0.564 | 0.661 | 0.659 |

| VL-MIA/DALL-E | | | | | | | | | | | | |
|---|---|---|---|---|---|---|---|---|---|---|---|---|
| **Metric** | | **LLaVA** | | | | **MiniGPT-4** | | | | **LLaMA Adapter** | | |
| | | img | inst | desp | inst+desp | img | inst | desp | inst+desp | inst | desp | inst+desp |
| Perplexity* | | N/A | 0.338 | 0.564 | 0.448 | N/A | 0.356 | 0.517 | 0.421 | 0.491 | 0.577 | 0.506 |
| Min_0% Prob* | | N/A | 0.482 | 0.559 | 0.482 | N/A | 0.422 | 0.494 | 0.421 | 0.448 | 0.554 | 0.448 |
| Min_10% Prob* | | N/A | 0.482 | 0.563 | 0.425 | N/A | 0.422 | 0.495 | 0.462 | 0.447 | 0.556 | 0.455 |
| Min_20% Prob* | | N/A | 0.434 | 0.559 | 0.353 | N/A | 0.462 | 0.501 | 0.401 | 0.560 | 0.460 | 0.456 |
| Aug_KL | | 0.408 | 0.463 | 0.505 | 0.489 | 0.396 | 0.421 | 0.460 | 0.446 | 0.474 | 0.489 | 0.476 |
| Max_Prob_Gap | | 0.529 | 0.575 | **0.597** | 0.602 | 0.527 | 0.407 | 0.505 | 0.490 | 0.518 | 0.553 | 0.555 |
| ModRényi* | $\alpha = 0.5$ | N/A | 0.360 | 0.560 | 0.523 | N/A | 0.399 | **0.518** | 0.465 | 0.479 | 0.580 | 0.546 |
| | $\alpha = 1$ | N/A | 0.342 | 0.560 | 0.425 | N/A | 0.372 | 0.516 | 0.450 | 0.478 | 0.576 | 0.489 |
| | $\alpha = 2$ | N/A | 0.384 | 0.561 | 0.536 | N/A | 0.416 | **0.518** | 0.477 | 0.486 | **0.581** | 0.559 |
| Rényi ($\alpha = 0.5$) | Max_0% | 0.537 | 0.597 | 0.563 | 0.598 | 0.518 | **0.496** | 0.493 | **0.497** | **0.625** | 0.503 | **0.624** |
| | Max_10% | 0.622 | 0.597 | 0.563 | **0.648** | 0.563 | **0.496** | 0.504 | 0.482 | 0.573 | 0.516 | 0.573 |
| | Max_100% | 0.421 | 0.604 | 0.575 | 0.582 | 0.528 | 0.448 | 0.504 | 0.481 | 0.511 | 0.552 | 0.531 |
| Rényi ($\alpha = 1$) | Max_0% | 0.549 | 0.569 | 0.551 | 0.576 | 0.523 | 0.477 | 0.497 | 0.486 | 0.598 | 0.522 | 0.597 |
| | Max_10% | 0.667 | 0.569 | 0.558 | 0.586 | 0.555 | 0.477 | 0.512 | 0.472 | 0.532 | 0.530 | 0.553 |
| | Max_100% | 0.469 | **0.637** | 0.564 | 0.584 | 0.548 | 0.428 | 0.517 | 0.477 | 0.519 | 0.555 | 0.532 |
| Rényi ($\alpha = 2$) | Max_0% | 0.591 | 0.549 | 0.545 | 0.558 | 0.524 | 0.401 | 0.489 | 0.445 | 0.504 | 0.529 | 0.503 |
| | Max_10% | **0.707** | 0.549 | 0.553 | 0.575 | 0.548 | 0.401 | 0.503 | 0.428 | 0.526 | 0.534 | 0.528 |
| | Max_100% | 0.526 | 0.606 | 0.560 | 0.576 | 0.548 | 0.406 | **0.518** | 0.476 | 0.509 | 0.556 | 0.530 |
| Rényi ($\alpha = \infty$) | Max_0% | 0.623 | 0.559 | 0.559 | 0.567 | 0.534 | 0.386 | 0.494 | 0.439 | 0.461 | 0.554 | 0.460 |
| | Max_10% | 0.699 | 0.559 | 0.563 | 0.580 | 0.555 | 0.386 | 0.495 | 0.416 | 0.510 | 0.556 | 0.515 |
| | Max_100% | 0.545 | 0.587 | 0.564 | 0.577 | 0.550 | 0.394 | 0.517 | 0.473 | 0.506 | 0.577 | 0.530 |

**MIA on VLLM instruction-tuning texts** We detect whether individual description texts appear in the VLLMs instruction-tuning. We use the description text dataset VL-MIA/Text of lengths (32, 64), constructed in Section 4. We present our results in the first column of Table 3. We observe that target-based MIA methods are significantly better than target-free ones, confirming our hypothesis.

**MIA on LLM pre-training texts.** We use the WikiMIA benchmark [52], which leverages the Wikipedia timestamp to separate the early Wiki data as the member data, and recent Wiki data as the non-member data. The early Wiki data are used in various LLMs pre-training [60]. We use WikiMIA of different lengths (32, 64, 128, 256), and expect the membership of longer sequences will be more easily identified. We present our results in the second column of Table 3. We observe that on LLaVA, our target-free MIA methods on large $\alpha$ consistently outperform target-based MIA methods, which

again confirms our hypothesis since the base LLM model of LLaVA has full parameter fine-tuning from LLaMA-2 and thus changed a lot.

## 6.4 Image MIA on GPT-4

In this section, we demonstrate the feasibility of image MIAs on the closed-source model GPT-4. Our experiments use two image datasets: VL-MIA/Flickr and VL-MIA/DALL-E, detailed in Section 4. We choose GPT-4-vision-preview API, which was trained in 2023 and likely does not see the member data in either dataset. We randomly select 200 images per dataset and prompt GPT-4 to describe them in 64 words. We then apply MIAs based on the generated descriptions. Since GPT-4 can only provide the top-five probabilities at each token position, we can not directly use the proposed `MaxRényi-K%` that requires the whole probability distribution. To address this issue, we assume the size of the entire token set is 32000 and the probability of the remaining 31995 tokens are uniformly distributed. The AUC results are present in Table 4. We omit the result of perplexity and `Min-K%` since they are equivalent to `MaxRényi-K%` with $\alpha = \infty$ in the greedy-generated setting, as discussed in Section 5.2. Surprisingly, we observe that in VL-MIA/DALL-E, the best-performed

Table 4: **Image MIA on GPT-4**.

| Metric | | VL-MIA/ DALL-E | VL-MIA/ Flickr |
|---|---|---|---|
| Perplexity/zlib* | | 0.807 | 0.520 |
| Max_Prob_Gap | | 0.516 | 0.486 |
| | Max_0% | 0.697 | 0.571 |
| Rényi ($\alpha = 0.5$) | Max_10% | 0.749 | 0.604 |
| | Max_100% | **0.815** | 0.605 |
| | Max_0% | 0.688 | 0.572 |
| Rényi ($\alpha = 1$) | Max_10% | 0.747 | 0.591 |
| | Max_100% | 0.790 | **0.630** |
| | Max_0% | 0.678 | 0.572 |
| Rényi ($\alpha = 2$) | Max_10% | 0.723 | 0.574 |
| | Max_100% | 0.786 | 0.595 |
| | Max_0% | 0.685 | 0.561 |
| Rényi ($\alpha = \infty$) | Max_10% | 0.708 | 0.549 |
| | Max_100% | 0.781 | 0.583 |

method `MaxRényi-K%` ($\alpha = 0.5$) can achieve an AUC of 0.815. This indicates a high level of effectiveness for MIAs on GPT-4, demonstrating the potential risks of privacy leakage even with closed-source models.

## 6.5 Ablation study

**Does the length of description affect the image MIA performance?** We conduct ablation experiments on LLaVA 1.5 targeting the length of generated description texts with `MaxRényi-10%`. In the generation stage, we restrict the `max_new_tokens` parameter of the generate function to (32, 64, 128, 256) to obtain description slices of different lengths. As presented in Figure 2a, when the length of the description increases, the AUC of the MIA becomes higher and enters a plateau when `max_new_tokens` reaches 128. This may be because a shorter text contains insufficient information about the image, and in an excessively long text, words generated later tend to be more generic and not closely related to the image, thereby contributing less to the discriminative information that helps discern the membership.

**Can we still detect corrupted member images?** The motivation is to detect whether sensitive images are inappropriately used in VLLM's training even when the images at hand may get corrupted. We leverage ImageNet-C [21] to generate corrupted versions of member data in VL-MIA/Flickr: `Snow`, `Brightness`, `JPEG`, and `Motion_Blur`, with the parameters in Table 8. The corrupted examples and corresponding model output generations are demonstrated in Appendix C Table 17 and Table 18. We take `MaxRényi-K%` ($\alpha = 0.5$) as the attacker and the results of LLaVA are presented in Figure 2b. Corrupted member images make MIAs more difficult, but can still be detected successfully. We also observe that reducing model quality (`JPEG`) or adding blur (`Motion_Blur`) degrade MIA performance more than changing the base parameter (`Brightness`) or overlaying texture (`Snow`).

**Can we use different instructions?** We conduct image MIAs on VL-MIA/Flickr with LLaVA through three different instruction texts: "Describe this image concisely.", "Please introduce this painting.", and "Tell me about this image.". We present our results in Table 14 of the appendix. Our pipeline successfully detects member images on every instruction, which indicates robustness across different instruction texts.

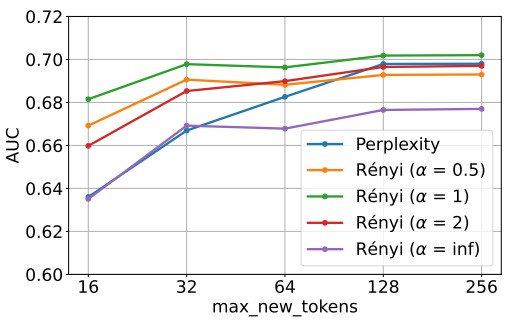

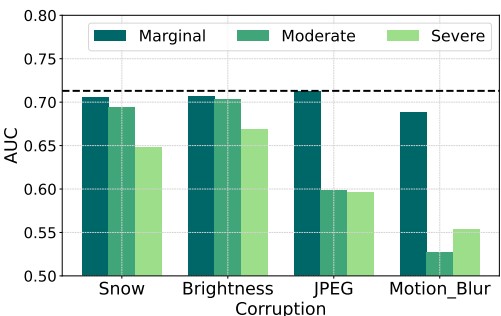

(a) Ablation study on `max_new_tokens`.

(b) `MaxRényi-K%` on corrupted images.

Figure 2: **Ablation study** (a) on `max_new_tokens` with `MaxRényi-10%`. Allowing VLLMs to generate longer descriptions can increase the AUC of "desp" slices, but we encounter a plateau when `max_new_tokens` equals 128. (b) on image MIAs against corrupted versions of VL-MIA/Flickr with `MaxRényi-K%` ($\alpha = 0.5$). Three levels of corruption are applied to the images: Marginal, Moderate, and Severe. The dotted line indicates the AUC on raw images without corruption.

# 7   Conclusion

In this work, we take an initial step towards detecting training data in VLLMs. Specifically, we construct a comprehensive dataset to perform MIAs on both image and text modalities. Additionally, we uncover a new pipeline for conducting MIA on VLLMs cross-modally and propose a novel method based on Rényi entropy. We believe that our work paves the way for advancing MIA techniques and, consequently, enhancing privacy protection in large foundation models.

# Acknowledgements

This work was carried out when Zhan Li and Yihang Chen were interns in the EPFL LIONS group. This work was supported by Hasler Foundation Program: Hasler Responsible AI (project number 21043). This research was sponsored by the Army Research Office and was accomplished under Grant Number W911NF-24-1-0048. This work was supported by the Swiss National Science Foundation (SNSF) under grant number 200021_205011.

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

# Contents of the appendix

The Appendix is organized as follows:

- In Appendix A, we provide some supplementary of the main body.

- Our additional experiment results are presented in Appendix B.

- A preview of some examples of VL-MIA can be found in Appendix C.

- We give broader impacts of this work in Appendix F.

- We state the limitation of our work in Appendix G.

## A  Supplementaries to the main text

### A.1  Detailed notation

We summarize the notations of this work in Table 5.

Table 5: Main notations.

| Notation | Meaning |
|---|---|
| $\mathcal{V}$ | The token set of a VLLM |
| $\mathbf{A}_{\text{image}}$ | Membership inference attacker for image |
| $\mathbf{A}_{\text{des}}$ | Membership inference attacker for text |
| $\theta$ | Model parameters |
| $X_{\text{des}}$, $Z$ | Text description and image to be detected |
| $X_{\text{ins}}$ | Text instruction, e.g., "Describe this image" |
| $X'_{\text{des}}$ | Generated text description |
| $X_{\text{ept}}$ | Empty instruction |
| $Z_{\text{ept}}$ | All-black image |
| $p$ | Some probability distribution |
| $p^{(i)}$ | The next-token probability distribution at position $i$ |
| $p_j$ | The probability corresponding the $j$-th token in $\mathcal{V}$ |
| $H_\alpha(p)$ | Rényi entropy of order $\alpha$ |
| $\overline{H}_\alpha(p)$ | Linearized Rényi entropy of order $\alpha$ |
| $\overline{H}_\alpha(p, y)$ | Modified Rényi entropy of order $\alpha$ with target $y$ |

### A.2  Explanation and visualization of slices

We give an example of the different slices in the MiniGPT-4 prompt in Figure 3.

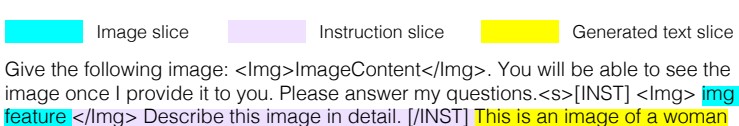

Figure 3: Different slices in the prompt.

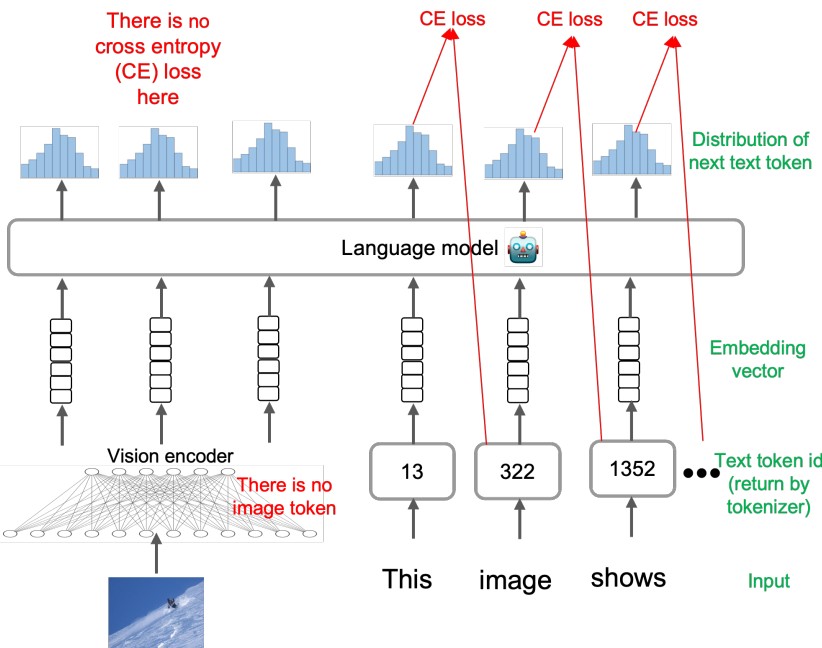

Figure 4: A schematic of VLLM.

As shown in Figure 4, the VLLMs consist of a vision encoder, a text tokenizer, and a language model. The output of the vision encoder (image embedding) have the same embedding dimension $d$ as the text token embedding.

When feeding the VLLMs with an image and the instruction, a vision encoder transforms this image to $L_1$ hidden embeddings of dimension $d$, denoted by $E_{\text{image}} \in \mathbb{R}^{d \times L_1}$. The text tokenizer first tokenizes the instruction into $L_2$ tokens, and then looks up the embedding matrix to get its $d$-dimensional embedding $E_{\text{ins}} \in \mathbb{R}^{d \times L_2}$. The image embedding and the instruction embedding are concatenated as $E_{\text{img}-\text{ins}} = (E_{\text{image}}, E_{\text{ins}}) \in \mathbb{R}^{d \times (L_1 + L_2)}$, which are then fed into a language model to perform next token prediction. The cross-entropy loss (CE loss) is calculated based on the predicted token id and the ground truth token id on the text tokens.

We can see that in this process, image embeddings are directly obtained from the vision encoder and there are no image tokens. There are no causal relations between consecutive image embeddings as well. Threfore, as we stated in Section 1, target-based MIA that requires token ids cannot be directly applied.

### A.3 Datasets construction prompts

We present the instruction prompts we use to construct the dataset in Section 4 in Table 6.

Table 6: Different prompts we use for dataset construction.

| Dataset | Prompt | Model |
|---|---|---|
| VL-MIA/DALL-E | {original caption in member data} | dall-e-2 |
| VL-MIA/Text for MiniGPT-4 | Provide a caption for this image, beginning with "The image". | gpt-4-vision-preview |
| VL-MIA/Text for LLaVA 1.5 | {original question in member data} | gpt-4-vision-preview |

### A.4 Additional experimental information

The versions, base models and training datasets of target VLLMs are listed in Table 7. We run experiments on a single NVIDIA A100 80GB GPU, where the image MIA costs less than 2 hours for one model.

In our experiments on description text MIA, we do not detect the member text during the VLLM image-text pre-training stage (the 3rd row) because the captions used in pre-training are usually fewer than 10 words and do not contain sufficient information. Therefore, we only detect member text during the VLLM instruction tuning stage.

Table 7: Model details used in this work.

| Model | Mini-GPT4 | LLaVA 1.5 | LLaMA Adapter v2.1 |
|---|---|---|---|
| Base LLM | Llama 2 Chat 7B | Vicuna v1.5 7B | LLaMA 7B |
| Vision processor | BLIP2/eva_vit_g[5] | CLIP-ViT-L | CLIP-ViT-L |
| Image-text pretraining | BLIP_LAION_CCS | BLIP_LAION_CCS | Image-Text-V1[6] |
| Instruction tuning | CC_SBU_Align | LLaVA v1.5 mix665k[7] | GPT4LLM, LLaVA Instruct 150k, VQAv2 |

### A.5 Parameters for image corruption

We utilize the code[8] from ImageNet-C [21] to corrupt member images. The related analysis can be found in Section 6.5. Here we provide the corruption parameters in the code in Table 8.

Table 8: Values for corruption parameter `c` in ImageNet-C code.

| | Brightness | Motion_Blur | Snow | JPEG |
|---|---|---|---|---|
| Marginal | 0.3 | (60,10) | (0.3,0.3,1.25,0.65,14,12,0.8) | 10 |
| Moderate | 0.5 | (90,20) | (0.4,0.4,1.5,0.8,17,15,0.6) | 4 |
| Severe | 0.7 | (120,30) | (0.5,0.5,2,0.95,19,18,0.45) | 2 |

## B  Additional experiments

### B.1 Text MIA: Complete results

In addition to Section 6.3, we present our complete Text MIA results on LLaVA, MiniGPT-4, and LLaMA-Adapter in Table 9 and Table 10. We find that no current method can effectively detect LLM pre-training text corpus from MiniGPT-4, even the AUC on 256-length Wiki text is only 0.6. Apart from that, we note that on the LLaMA-Adapter, target-based methods perform similarly or better than target-free ones. In the following, we explain this phenomenon by examining the training pipelines of two different models.

The wiki data is used in the pre-training stage of LLaMA 7b, which is further fine-tuned to LLaMA 7b Chat. The base MML of LLAVA is fine-tuned from Vicuna 7b v1.5, which is further fine-tuned from LLaMA 7b Chat. Therefore, the LLM model of LLAVA has changed a lot since wiki data's last usage, and target-free methods are preferable here. In contrast, LLaMA-Adapter is parameter-efficiently fine-tuned from LLaMA 7b, and the model's parameters have not changed much since the wiki's data last usage, and the target-based MIA methods can still retain their performance. In addition, to

---

[5]BLIP2/eva_vit_g uses MS COCO as one of the pre-training datasets.

[6]A concatenation of LAION400M, COYO, MMC4, SBU, Conceptual Captions, and COCO.

[7]Including LLaVA Instruct 150k and MS COCO.

[8]https://github.com/hendrycks/robustness/blob/master/ImageNet-C/create_c/make_cifar_c.py

detect instruction-tuning texts in VLLMs, the model has just repeatedly seen the member data, and the target-based ones are significantly superior to target-free ones. Overall, the phenomena can be explained by our hypothesis.

Table 9: **MIA on VLLM instruction-tuning texts**. We present the MIA on VLLM instruction-tuning texts with description length (32, 64). Complete results of Table 3.

| Metric | | **LLaVA** | | **MiniGPT-4** | | **LLaMA Adapter** | |
|---|---|---|---|---|---|---|---|
| | | 32 | 64 | 32 | 64 | 32 | 64 |
| Perplexity* | | 0.779 | 0.988 | **0.952** | 0.993 | 0.857 | 0.994 |
| Perplexity/zlib* | | 0.609 | 0.986 | 0.851 | 0.950 | 0.743 | 0.995 |
| Perplexity/lowercase* | | **0.962** | 0.977 | 0.678 | 0.801 | **0.943** | 0.958 |
| Min_0% Prob* | | 0.522 | 0.522 | 0.726 | 0.771 | 0.630 | 0.672 |
| Min_10% Prob* | | 0.461 | 0.883 | 0.802 | 0.931 | 0.708 | 0.932 |
| Min_20% Prob* | | 0.603 | 0.980 | 0.881 | 0.991 | 0.782 | 0.988 |
| Max_Prob_Gap | | 0.461 | 0.545 | 0.643 | 0.787 | 0.459 | 0.591 |
| | $\alpha = 0.5$ | 0.809 | 0.979 | 0.915 | 0.986 | 0.844 | 0.987 |
| ModRényi* | $\alpha = 1$ | 0.808 | **0.993** | **0.952** | **0.994** | 0.876 | **0.996** |
| | $\alpha = 2$ | 0.779 | 0.963 | 0.892 | 0.978 | 0.815 | 0.976 |
| | Max_0% | 0.506 | 0.514 | 0.533 | 0.511 | 0.517 | 0.510 |
| Rényi ($\alpha = 0.5$) | Max_10% | 0.458 | 0.776 | 0.368 | 0.512 | 0.505 | 0.836 |
| | Max_100% | 0.564 | 0.835 | 0.805 | 0.945 | 0.570 | 0.835 |
| | Max_0% | 0.552 | 0.579 | 0.528 | 0.508 | 0.590 | 0.603 |
| Rényi ($\alpha = 1$) | Max_10% | 0.566 | 0.809 | 0.566 | 0.781 | 0.591 | 0.828 |
| | Max_100% | 0.554 | 0.750 | 0.756 | 0.911 | 0.561 | 0.762 |
| | Max_0% | 0.589 | 0.625 | 0.648 | 0.690 | 0.615 | 0.635 |
| Rényi ($\alpha = 2$) | Max_10% | 0.607 | 0.787 | 0.681 | 0.843 | 0.610 | 0.795 |
| | Max_100% | 0.553 | 0.709 | 0.755 | 0.902 | 0.554 | 0.716 |
| | Max_0% | 0.600 | 0.638 | 0.679 | 0.765 | 0.612 | 0.616 |
| Rényi ($\alpha = \infty$) | Max_10% | 0.618 | 0.763 | 0.688 | 0.843 | 0.615 | 0.755 |
| | Max_100% | 0.557 | 0.694 | 0.747 | 0.896 | 0.549 | 0.695 |

Table 10: MIA on LLM pre-training texts. We evaluate on WikiMIA of lengths (32, 64, 128, 256). Complete results of Table 3.

| Metric | | **LLaVA** | | | | **MiniGPT-4** | | | | **LLaMA Adapter** | | | |
|---|---|---|---|---|---|---|---|---|---|---|---|---|---|
| | | 32 | 64 | 128 | 256 | 32 | 64 | 128 | 256 | 32 | 64 | 128 | 256 |
| Perplexity* | | 0.542 | 0.505 | 0.553 | 0.582 | 0.523 | 0.484 | 0.551 | 0.588 | 0.614 | 0.583 | 0.643 | 0.666 |
| Perplexity/zlib* | | 0.56 | 0.537 | 0.581 | 0.603 | 0.545 | **0.515** | **0.575** | **0.607** | **0.624** | 0.606 | 0.661 | 0.689 |
| Perplexity/lowercase* | | 0.493 | 0.518 | 0.503 | 0.583 | 0.498 | 0.506 | 0.531 | 0.605 | 0.574 | 0.573 | 0.571 | 0.594 |
| Min_0% Prob* | | 0.455 | 0.451 | 0.425 | 0.448 | 0.430 | 0.427 | 0.397 | 0.393 | 0.535 | 0.569 | 0.595 | 0.513 |
| Min_10% Prob* | | 0.468 | 0.487 | 0.526 | 0.534 | 0.470 | 0.485 | 0.545 | 0.593 | 0.607 | **0.615** | **0.672** | 0.661 |
| Max_Prob_Gap | | 0.574 | 0.544 | 0.565 | 0.629 | 0.498 | 0.48 | 0.527 | 0.572 | 0.587 | 0.56 | 0.604 | **0.698** |
| | $\alpha = 0.5$ | 0.557 | 0.500 | 0.536 | 0.567 | 0.557 | 0.500 | 0.540 | 0.578 | 0.576 | 0.539 | 0.600 | 0.648 |
| ModRényi* | $\alpha = 1$ | 0.544 | 0.503 | 0.546 | 0.567 | 0.531 | 0.489 | 0.552 | 0.587 | 0.613 | 0.583 | 0.642 | 0.66 |
| | $\alpha = 2$ | 0.559 | 0.497 | 0.529 | 0.560 | **0.565** | 0.504 | 0.537 | 0.571 | 0.569 | 0.531 | 0.591 | 0.653 |
| | Max_0% | 0.541 | 0.515 | 0.489 | 0.571 | 0.49 | 0.495 | 0.505 | 0.502 | 0.500 | 0.464 | 0.497 | 0.564 |
| Rényi ($\alpha = 0.5$) | Max_10% | 0.518 | 0.525 | 0.606 | 0.65 | 0.501 | 0.464 | 0.536 | 0.586 | 0.511 | 0.51 | 0.625 | 0.693 |
| | Max_100% | 0.555 | 0.531 | 0.600 | 0.631 | 0.511 | 0.491 | 0.547 | 0.603 | 0.553 | 0.541 | 0.626 | 0.678 |
| | Max_0% | 0.566 | 0.571 | 0.603 | 0.668 | 0.476 | 0.472 | 0.499 | 0.568 | 0.562 | 0.576 | 0.615 | 0.596 |
| Rényi ($\alpha = 1$) | Max_10% | 0.553 | 0.541 | 0.623 | 0.65 | 0.500 | 0.474 | 0.556 | 0.584 | 0.549 | 0.545 | 0.65 | 0.685 |
| | Max_100% | 0.544 | 0.523 | 0.588 | 0.621 | 0.499 | 0.484 | 0.538 | 0.576 | 0.553 | 0.546 | 0.622 | 0.686 |
| | Max_0% | 0.594 | 0.606 | 0.659 | 0.657 | 0.498 | 0.485 | 0.568 | 0.600 | 0.590 | 0.590 | 0.636 | 0.627 |
| Rényi ($\alpha = 2$) | Max_10% | 0.583 | 0.556 | 0.629 | 0.663 | 0.495 | 0.478 | 0.555 | 0.592 | 0.576 | 0.568 | 0.649 | 0.692 |
| | Max_100% | 0.544 | 0.523 | 0.583 | 0.622 | 0.492 | 0.480 | 0.529 | 0.572 | 0.555 | 0.55 | 0.613 | 0.685 |
| | Max_0% | **0.607** | **0.615** | **0.688** | **0.669** | 0.514 | 0.497 | 0.537 | 0.591 | 0.581 | 0.57 | 0.638 | 0.579 |
| Rényi ($\alpha = \infty$) | Max_10% | 0.586 | 0.548 | 0.627 | 0.667 | 0.466 | 0.462 | 0.542 | 0.598 | 0.574 | 0.57 | 0.643 | 0.694 |
| | Max_100% | 0.546 | 0.527 | 0.584 | 0.634 | 0.488 | 0.477 | 0.527 | 0.572 | 0.557 | 0.551 | 0.612 | 0.689 |

## B.2 Abalation on `MaxRényi` and `MinRényi`

Given a sequence of entropies calculated on each position, it would be natural to ask whether the higher percentile or lower percentile should be used to determine the score of the whole sequence for MIA. In the main paper, we use `max`, which coincides with `Min-K%` [52] under some special cases. We provide more evidence of the advantages of `max` or `min`. We define `MinRényi-K%` as similar to `MaxRényi-K%`, but consider the smallest K% entropies in the sequence $X$.

We conduct the pre-training text MIA on LLAVA. We also use the WiKiMIA [52] benchmark of different lengths (32, 64, 128, 256). We set $K = 0, 10, 20$, since when $K = 100$, `MaxRényi-K%` coincides with `MinRényi-K%`. We observe that usually `MinRényi-K%` is slightly inferior to `MaxRényi-K%`. Therefore, we adopt `MaxRényi-K%` in our main paper.

Table 11: **MIA on LLM pre-training texts**. We evaluate on WikiMIA of lengths (32, 64, 128, 256) on LLaVA. We compare MaxRényi with MinRényi.

| Metric | | MaxRényi | | | | MinRényi | | | |
|---|---|---|---|---|---|---|---|---|---|
| | | 32 | 64 | 128 | 256 | 32 | 64 | 128 | 256 |
| Rényi ($\alpha = 0.5$) | 0% | 0.541 | 0.515 | 0.489 | 0.571 | 0.557 | 0.515 | 0.488 | 0.553 |
| | 10% | 0.518 | 0.525 | 0.606 | 0.65 | 0.569 | 0.525 | 0.605 | **0.647** |
| | 20% | 0.530 | 0.530 | 0.616 | 0.642 | 0.558 | 0.530 | 0.616 | 0.640 |
| Rényi ($\alpha = 1$) | 0% | 0.566 | 0.571 | 0.603 | 0.668 | 0.563 | 0.570 | 0.603 | 0.521 |
| | 10% | 0.553 | 0.541 | 0.623 | 0.65 | **0.573** | 0.541 | 0.622 | **0.647** |
| | 20% | 0.548 | 0.527 | 0.611 | 0.630 | 0.564 | 0.527 | 0.611 | 0.631 |
| Rényi ($\alpha = 2$) | 0% | 0.594 | 0.606 | 0.659 | 0.657 | 0.563 | 0.605 | **0.658** | 0.521 |
| | 10% | 0.583 | 0.556 | 0.629 | 0.663 | 0.572 | 0.555 | 0.628 | 0.645 |
| | 20% | 0.559 | 0.527 | 0.611 | 0.642 | 0.565 | 0.527 | 0.611 | 0.631 |
| Rényi ($\alpha = \infty$) | 0% | **0.607** | **0.615** | **0.688** | **0.669** | 0.563 | **0.615** | **0.687** | 0.520 |
| | 10% | 0.586 | 0.548 | 0.627 | 0.667 | 0.572 | 0.548 | 0.626 | 0.645 |
| | 20% | 0.552 | 0.527 | 0.608 | 0.651 | 0.565 | 0.527 | 0.608 | 0.531 |

## B.3 Ablation on extended dataset

In this part, we extend VL-MIA/Flickr and VL-MIA/Text size to 2000, which includes 1000 members and 1000 non-members. From Table 12 and Table 13, we see a similar trend with small-size (600 samples) VL-MIA. Considering this computing source, our main experiments are conducted on small-size datasets.

## B.4 Different instruction texts

We conduct image MIA on VL-MIA/Flickr with LLaVA through three different instruction texts: "Describe this image concisely.", "Please introduce this painting.", and "Tell me about this image.". We present our results in Table 14 of the appendix. Our pipeline successfully detects member images on every instruction, which indicates robustness across different instruction texts.

## C Dataset examples

In Table 16, we give some examples of our proposed VL-MIA datasets. In Table 17 and Table 18, we show the corrupted images used in the ablation stuidies (Section 6.5) and the model's descriptions of these corrupted images. In Figure 5, we present some examples of our VL-MIA/Synthetic dataset.

## D TPR at 5% FPR

In this section, we provide the True Positive Rate (TPR) at 5% False Positive Rate (FPR) results as supplementary evaluation, as presented in Table 15. We can see a similar trend in Table 2.

Table 12: **Image MIA on extended VL-MIA/Flickr**. AUC results on VL-MIA/Flickr of size 2000 under our pipeline.

| Metric | | LLaVA | | | | MiniGPT-4 | | | | LLaMA Adapter | | |
|---|---|---|---|---|---|---|---|---|---|---|---|---|
| | | img | inst | desp | inst+desp | img | inst | desp | inst+desp | inst | desp | inst+desp |
| Perplexity* | | N/A | 0.365 | 0.665 | 0.561 | N/A | 0.556 | 0.616 | 0.578 | 0.287 | 0.645 | 0.314 |
| Min_0% Prob* | | N/A | 0.353 | 0.597 | 0.353 | N/A | 0.366 | 0.563 | 0.366 | 0.394 | 0.547 | 0.394 |
| Min_10% Prob* | | N/A | 0.353 | 0.606 | 0.336 | N/A | 0.366 | 0.586 | 0.383 | 0.352 | 0.565 | 0.339 |
| Min_20% Prob* | | N/A | 0.335 | 0.619 | 0.345 | N/A | 0.437 | 0.601 | 0.453 | 0.339 | 0.584 | 0.316 |
| Aug_KL | | 0.586 | 0.535 | 0.483 | 0.504 | 0.446 | 0.498 | 0.489 | 0.491 | 0.480 | 0.476 | 0.480 |
| Max_Prob_Gap | | 0.602 | 0.516 | 0.639 | 0.637 | **0.642** | 0.648 | 0.578 | 0.590 | 0.562 | 0.668 | 0.676 |
| ModRényi* | $\alpha = 0.5$ | N/A | 0.528 | 0.658 | 0.681 | N/A | 0.533 | 0.613 | 0.614 | 0.336 | 0.644 | 0.455 |
| | $\alpha = 1$ | N/A | 0.379 | 0.656 | 0.513 | N/A | 0.505 | 0.612 | 0.510 | 0.295 | 0.624 | 0.311 |
| | $\alpha = 2$ | N/A | 0.528 | 0.659 | 0.680 | N/A | 0.528 | 0.612 | 0.615 | 0.343 | 0.651 | 0.500 |
| Rényi ($\alpha = 0.5$) | Max_0% | 0.559 | 0.647 | 0.656 | 0.648 | 0.493 | 0.569 | 0.566 | 0.571 | 0.612 | 0.612 | 0.611 |
| | Max_10% | 0.561 | 0.647 | 0.659 | 0.675 | 0.525 | 0.569 | 0.587 | 0.591 | 0.599 | 0.621 | 0.706 |
| | Max_100% | **0.711** | **0.685** | **0.687** | **0.695** | 0.624 | 0.685 | 0.617 | 0.655 | **0.713** | 0.677 | 0.731 |
| Rényi ($\alpha = 1$) | Max_0% | 0.534 | 0.641 | 0.620 | 0.645 | 0.482 | 0.563 | 0.568 | 0.578 | 0.609 | 0.606 | 0.608 |
| | Max_10% | 0.572 | 0.641 | 0.633 | 0.658 | 0.516 | 0.563 | 0.593 | 0.611 | 0.633 | 0.622 | 0.685 |
| | Max_100% | 0.709 | 0.620 | 0.679 | 0.678 | 0.599 | 0.673 | **0.619** | 0.668 | 0.699 | **0.680** | **0.740** |
| Rényi ($\alpha = 2$) | Max_0% | 0.563 | 0.608 | 0.606 | 0.623 | 0.482 | 0.601 | 0.567 | 0.612 | 0.600 | 0.602 | 0.599 |
| | Max_10% | 0.583 | 0.608 | 0.617 | 0.636 | 0.511 | 0.601 | 0.591 | 0.649 | 0.670 | 0.619 | 0.720 |
| | Max_100% | 0.690 | 0.587 | 0.672 | 0.669 | 0.600 | **0.698** | 0.618 | **0.671** | 0.680 | 0.675 | 0.733 |
| Rényi ($\alpha = \infty$) | Max_0% | 0.554 | 0.579 | 0.597 | 0.604 | 0.488 | 0.594 | 0.563 | 0.608 | 0.564 | 0.547 | 0.564 |
| | Max_10% | 0.568 | 0.579 | 0.606 | 0.619 | 0.513 | 0.594 | 0.586 | 0.635 | 0.658 | 0.565 | 0.703 |
| | Max_100% | 0.676 | 0.568 | 0.665 | 0.660 | 0.604 | 0.682 | 0.616 | 0.668 | 0.661 | 0.645 | 0.720 |

Table 13: **Text MIA on extended VL-MIA/Text.** AUC results on LLaVA with 1000 members and 1000 non-members. We detect the text used in VLLM instruction tuning stage for text length equals to [32,64].

| Metric | | 32 | 64 |
|---|---|---|---|
| Perplexity* | | 0.784 | 0.986 |
| Perplexity/zlib* | | 0.626 | 0.988 |
| Perplexity/lowercase* | | **0.965** | 0.983 |
| Min_0% Prob* | | 0.525 | 0.527 |
| Min_10% Prob* | | 0.463 | 0.908 |
| Min_20% Prob* | | 0.598 | 0.982 |
| Max_Prob_Gap | | 0.465 | 0.562 |
| ModRényi* | $\alpha = 0.5$ | 0.809 | 0.977 |
| | $\alpha = 1$ | 0.811 | **0.992** |
| | $\alpha = 2$ | 0.780 | 0.961 |
| Rényi ($\alpha = 0.5$) | Max_0% | 0.504 | 0.510 |
| | Max_10% | 0.458 | 0.790 |
| | Max_100% | 0.573 | 0.847 |
| Rényi ($\alpha = 1$) | Max_0% | 0.544 | 0.572 |
| | Max_10% | 0.557 | 0.804 |
| | Max_100% | 0.555 | 0.762 |
| Rényi ($\alpha = 2$) | Max_0% | 0.577 | 0.608 |
| | Max_10% | 0.593 | 0.770 |
| | Max_100% | 0.554 | 0.720 |
| Rényi ($\alpha = \infty$) | Max_0% | 0.582 | 0.609 |
| | Max_10% | 0.595 | 0.742 |
| | Max_100% | 0.555 | 0.703 |

# E    VLLM pipelines

In this part, we investigate the pipelines of vLLMs, to explain why we have access to the text tokens, but only have access to the image embeddings instead of image tokens.

Table 14: **Image MIA**. AUC results on VL-MIA/Flickr with LLaVA 1.5 when we change the instruction text. "Describe" indicates "Describe this image concisely.", "Please" indicates "Please introduce this painting.", and "Tell" indicates "Tell me about this image.".

| Metric | | Describe | | | | Please | | | | Tell | | | |
|---|---|---|---|---|---|---|---|---|---|---|---|---|---|
| | | img | inst | desp | inst+desp | img | inst | desp | inst+desp | img | inst | desp | inst+desp |
| Perplexity* | | N/A | 0.378 | 0.667 | 0.559 | N/A | 0.379 | 0.671 | 0.549 | N/A | 0.362 | 0.662 | 0.530 |
| Min_0% Prob* | | N/A | 0.357 | 0.651 | 0.357 | N/A | 0.421 | 0.629 | 0.421 | N/A | 0.341 | 0.622 | 0.341 |
| Min_10% Prob* | | N/A | 0.357 | 0.669 | 0.390 | N/A | 0.421 | 0.656 | 0.414 | N/A | 0.341 | 0.646 | 0.367 |
| Min_20% Prob* | | N/A | 0.374 | 0.670 | 0.370 | N/A | 0.411 | 0.661 | 0.381 | N/A | 0.356 | 0.650 | 0.360 |
| Aug_KL | | 0.596 | 0.539 | 0.492 | 0.508 | 0.602 | 0.497 | 0.506 | 0.498 | 0.599 | 0.493 | 0.480 | 0.482 |
| Max_Prob_Gap | | 0.577 | 0.601 | 0.650 | 0.650 | 0.577 | 0.462 | 0.652 | 0.649 | 0.577 | 0.543 | 0.661 | 0.663 |
| ModRényi* | $\alpha = 0.5$ | N/A | 0.368 | 0.651 | 0.614 | N/A | 0.431 | 0.661 | 0.636 | N/A | 0.359 | 0.649 | 0.605 |
| | $\alpha = 1$ | N/A | 0.359 | 0.659 | 0.502 | N/A | 0.396 | 0.664 | 0.503 | N/A | 0.355 | 0.653 | 0.474 |
| | $\alpha = 2$ | N/A | 0.370 | 0.645 | 0.611 | N/A | 0.444 | 0.655 | 0.641 | N/A | 0.371 | 0.644 | 0.610 |
| Rényi ($\alpha = 0.5$) | Max_0% | 0.515 | 0.689 | 0.687 | 0.689 | 0.515 | **0.683** | 0.651 | 0.683 | 0.515 | 0.700 | 0.663 | 0.701 |
| | Max_10% | 0.557 | 0.689 | 0.691 | 0.719 | 0.557 | **0.683** | 0.663 | 0.666 | 0.557 | 0.700 | 0.672 | 0.723 |
| | Max_100% | **0.702** | **0.726** | **0.713** | 0.728 | **0.702** | 0.609 | **0.700** | 0.704 | 0.702 | **0.709** | **0.708** | **0.726** |
| Rényi ($\alpha = 1$) | Max_0% | 0.503 | 0.708 | 0.685 | 0.725 | 0.503 | 0.619 | 0.649 | 0.643 | 0.503 | 0.614 | 0.649 | 0.635 |
| | Max_10% | 0.623 | 0.708 | 0.698 | **0.743** | 0.623 | 0.619 | 0.670 | 0.707 | 0.623 | 0.614 | 0.671 | 0.699 |
| | Max_100% | 0.702 | 0.720 | 0.702 | 0.721 | 0.702 | 0.613 | 0.693 | 0.702 | 0.702 | 0.663 | 0.696 | 0.714 |
| Rényi ($\alpha = 2$) | Max_0% | 0.583 | 0.682 | 0.673 | 0.705 | 0.583 | 0.584 | 0.637 | 0.669 | 0.583 | 0.585 | 0.630 | 0.619 |
| | Max_10% | 0.621 | 0.682 | 0.685 | 0.725 | 0.621 | 0.584 | 0.660 | 0.670 | 0.621 | 0.585 | 0.655 | 0.672 |
| | Max_100% | 0.682 | 0.694 | 0.683 | 0.703 | 0.682 | 0.571 | 0.678 | 0.681 | 0.682 | 0.624 | 0.676 | 0.690 |
| Rényi ($\alpha = \infty$) | Max_0% | 0.588 | 0.646 | 0.651 | 0.674 | 0.588 | 0.636 | 0.629 | 0.666 | 0.588 | 0.578 | 0.622 | 0.603 |
| | Max_10% | 0.593 | 0.646 | 0.669 | 0.699 | 0.593 | 0.636 | 0.656 | 0.673 | 0.593 | 0.578 | 0.646 | 0.657 |
| | Max_100% | 0.669 | 0.673 | 0.667 | 0.687 | 0.669 | 0.539 | 0.671 | 0.667 | 0.669 | 0.608 | 0.662 | 0.675 |

The VLLMs consist of a vision encoder, a text tokenizer, and a language model. The vision encoder and the text tokenizer have the same embedding dimension $d$. When feeding the VLLMs with an image and the instruction, a vision encoder transforms this image to $L_1$ hidden embeddings of dimension $d$, denoted by $e_{\text{image}} \in \mathbb{R}^{d \times L_1}$. The text tokenizer first tokenizes the instruction into $L_2$ tokens, and then looks up the embedding matrix to get its $d$-dimensional embedding $e_{\text{ins}} \in \mathbb{R}^{d \times L_2}$. The image embedding and the instruction embedding are concatenated as $e_{\text{img}-\text{ins}} = (e_{\text{image}}, e_{\text{ins}}) \in \mathbb{R}^{d \times (L_1 + L_2)}$, which are then fed into a language model to generate a description that has $L_3$ tokens. The cross-entropy loss (CE loss) is calculated based on the predicted token id and the ground truth token id on the instruction and description tokens.

We can see that in this process, image embeddings are directly obtained from the vision encoder and there are no image tokens. Threfore, as we stated in Section 1, target-based MIAs that require token ids cannot be directly applied.

Furthermore, given the ouput logits of the shape $(L_1 + L_2 + L_3) \times |\mathcal{V}|$, where $\mathcal{V}$ is the vocabulary set, we can access the logits of the image by the slice $[0 : L_1]$, the logits of instruction by the slice $[L_1 : L_1 + L_2]$, and the logits of description by the slice $[L_1 + L_2 : L_1 + L_2 + L_3]$.

# F    Broader impacts

In this paper, we present the first multi-modalities MIA benchmark for VLLMs, and propose a novel metric `MaxRényi-K%` for MIA. We recognize that our research has significant implications for the safety and ethics of VLLMs and may lead to targeted MIA defense by developers. Nevertheless, our findings provide valuable insights into data contamination, which could contribute to the training data split. Additionally, our method empowers individuals to detect their private data within the training dataset, which is essential for ensuring data security. We believe our work can raise awareness about the importance of privacy protection in multi-modal language models.

# G    Limitation

The first limitation of this work is that the best methods on the pre-training dataset can only achieve an AUC of $0.688$. This is because the detection of pre-training data will become more challenging as the VLLMs are further fine-tuned. We believe increasing the performance for detecting pre-training data would be promising in the future. Secondly, the proposed `MaxRényi-K%` method requires access to the full probability distribution over the predicted tokens. Although we have demonstrated the

Table 15: **Image MIA**. TPR at 5% FPR results on VL-MIA under our pipeline. "img" indicates the logits slice corresponding to image embedding, "inst" indicates the instruction slice, "desp" the generated description slice, and "inst+desp" is the concatenation of the instruction slice and description slice. We use an asterisk * in superscript to indicate the target-based metric. **Bold** indicates the best AUC within each column and underline indicates the runner-up.

**VL-MIA/Flickr**

| Metric | | LLaVA | | | | MiniGPT-4 | | | | LLaMA Adapter | | |
|---|---|---|---|---|---|---|---|---|---|---|---|---|
| | | img | inst | desp | inst+desp | img | inst | desp | inst+desp | inst | desp | inst+desp |
| Perplexity* | | N/A | 0.007 | 0.130 | 0.070 | N/A | 0.000 | 0.067 | 0.073 | 0.017 | 0.157 | 0.023 |
| Min_0% Prob* | | N/A | 0.023 | 0.093 | 0.023 | N/A | 0.010 | 0.083 | 0.007 | 0.033 | 0.120 | 0.033 |
| Min_10% Prob* | | N/A | 0.023 | 0.083 | 0.013 | N/A | 0.010 | 0.103 | 0.003 | 0.020 | 0.097 | 0.017 |
| Min_20% Prob* | | N/A | 0.007 | 0.127 | 0.003 | N/A | 0.003 | 0.113 | 0.003 | 0.017 | 0.110 | 0.017 |
| Aug_KL | | 0.027 | 0.033 | 0.053 | 0.043 | 0.080 | 0.013 | 0.017 | 0.013 | 0.010 | 0.010 | 0.007 |
| Max_Prob_Gap | | 0.053 | 0.083 | 0.160 | 0.160 | 0.177 | 0.113 | 0.023 | 0.033 | 0.037 | 0.143 | 0.127 |
| | $\alpha = 0.5$ | N/A | 0.003 | 0.117 | 0.110 | N/A | 0.043 | 0.080 | 0.110 | 0.000 | 0.147 | 0.040 |
| ModRényi* | $\alpha = 1$ | N/A | 0.007 | 0.117 | 0.070 | N/A | 0.003 | 0.080 | 0.013 | 0.013 | 0.150 | 0.037 |
| | $\alpha = 2$ | N/A | 0.003 | 0.117 | 0.117 | N/A | 0.100 | 0.080 | 0.107 | 0.000 | 0.157 | 0.047 |
| | Max_0% | 0.047 | 0.203 | 0.083 | **0.200** | 0.033 | 0.053 | **0.133** | 0.053 | 0.063 | 0.053 | 0.063 |
| Rényi ($\alpha = 0.5$) | Max_10% | 0.110 | 0.203 | 0.063 | 0.140 | 0.050 | 0.053 | 0.130 | 0.117 | 0.040 | 0.067 | 0.100 |
| | Max_100% | 0.103 | **0.217** | 0.163 | 0.177 | **0.187** | 0.180 | 0.090 | **0.223** | 0.087 | 0.157 | **0.247** |
| | Max_0% | 0.053 | 0.133 | 0.120 | 0.167 | 0.027 | 0.050 | 0.110 | 0.073 | 0.077 | 0.047 | 0.080 |
| Rényi ($\alpha = 1$) | Max_10% | 0.087 | 0.133 | 0.070 | 0.120 | 0.070 | 0.050 | 0.130 | 0.130 | 0.043 | 0.057 | 0.080 |
| | Max_100% | 0.090 | 0.117 | 0.137 | 0.160 | 0.083 | 0.157 | 0.087 | 0.173 | 0.080 | **0.197** | 0.203 |
| | Max_0% | 0.070 | 0.113 | 0.093 | 0.150 | 0.053 | 0.167 | 0.090 | 0.183 | 0.097 | 0.060 | 0.107 |
| Rényi ($\alpha = 2$) | Max_10% | 0.083 | 0.113 | 0.093 | 0.103 | 0.050 | 0.167 | 0.103 | 0.220 | 0.103 | 0.073 | 0.157 |
| | Max_100% | 0.090 | 0.093 | **0.173** | 0.197 | 0.120 | **0.200** | 0.073 | 0.177 | 0.090 | **0.197** | 0.140 |
| | Max_0% | 0.097 | 0.097 | 0.093 | 0.170 | 0.050 | 0.163 | 0.083 | 0.163 | **0.133** | 0.120 | 0.140 |
| Rényi ($\alpha = \infty$) | Max_10% | **0.127** | 0.097 | 0.083 | 0.113 | 0.050 | 0.163 | 0.103 | 0.110 | 0.113 | 0.097 | 0.117 |
| | Max_100% | 0.087 | 0.113 | 0.130 | 0.167 | 0.123 | 0.170 | 0.067 | 0.150 | 0.073 | 0.157 | 0.190 |

**VL-MIA/DALL-E**

| Metric | | LLaVA | | | | MiniGPT-4 | | | | LLaMA Adapter | | |
|---|---|---|---|---|---|---|---|---|---|---|---|---|
| | | img | inst | desp | inst+desp | img | inst | desp | inst+desp | inst | desp | inst+desp |
| Perplexity* | | N/A | 0.027 | 0.081 | 0.057 | N/A | 0.014 | 0.034 | 0.034 | 0.030 | **0.081** | 0.051 |
| Min_0% Prob* | | N/A | 0.081 | 0.051 | 0.081 | N/A | 0.054 | 0.030 | 0.051 | 0.037 | 0.051 | 0.037 |
| Min_0% Prob* | | N/A | 0.081 | 0.068 | 0.041 | N/A | 0.054 | 0.020 | 0.057 | 0.020 | 0.030 | 0.020 |
| Min_20% Prob* | | N/A | 0.064 | 0.071 | 0.030 | N/A | 0.064 | 0.020 | 0.054 | 0.020 | 0.034 | 0.017 |
| Aug_KL | | 0.020 | 0.081 | 0.037 | 0.051 | 0.014 | 0.064 | 0.030 | 0.037 | 0.030 | 0.034 | 0.027 |
| Max_Prob_Gap | | 0.037 | 0.108 | 0.085 | 0.064 | 0.051 | 0.037 | 0.037 | 0.030 | 0.047 | 0.074 | 0.071 |
| | $\alpha = 0.5$ | N/A | 0.020 | 0.088 | 0.041 | N/A | 0.024 | 0.030 | 0.027 | 0.037 | **0.081** | 0.064 |
| ModRényi* | $\alpha = 1$ | N/A | 0.034 | 0.095 | 0.057 | N/A | 0.020 | 0.027 | **0.061** | 0.041 | 0.074 | 0.041 |
| | $\alpha = 2$ | N/A | 0.024 | 0.088 | 0.054 | N/A | 0.037 | 0.030 | 0.030 | 0.034 | 0.068 | 0.061 |
| | Max_0% | 0.081 | 0.088 | 0.047 | 0.085 | 0.061 | 0.061 | 0.020 | **0.061** | **0.112** | 0.051 | 0.112 |
| Rényi ($\alpha = 0.5$) | Max_10% | 0.152 | 0.088 | 0.064 | 0.095 | 0.068 | 0.061 | 0.017 | **0.061** | **0.162** | 0.061 | 0.044 |
| | Max_100% | 0.003 | 0.098 | 0.095 | 0.081 | 0.088 | 0.064 | 0.017 | 0.020 | 0.030 | 0.047 | 0.034 |
| | Max_0% | 0.091 | 0.101 | 0.064 | 0.044 | 0.054 | 0.037 | 0.027 | 0.044 | 0.088 | 0.054 | 0.081 |
| Rényi ($\alpha = 1$) | Max_10% | **0.220** | 0.101 | 0.061 | 0.064 | 0.074 | 0.037 | 0.014 | 0.037 | 0.071 | 0.041 | 0.051 |
| | Max_100% | 0.003 | 0.101 | 0.081 | 0.081 | 0.061 | **0.078** | 0.027 | 0.017 | 0.027 | 0.051 | 0.034 |
| | Max_0% | 0.118 | 0.091 | 0.051 | 0.054 | 0.051 | 0.030 | 0.030 | 0.027 | 0.051 | 0.054 | 0.047 |
| Rényi ($\alpha = 2$) | Max_10% | 0.172 | 0.091 | 0.051 | 0.068 | 0.071 | 0.030 | 0.020 | 0.017 | 0.034 | 0.037 | 0.057 |
| | Max_100% | 0.017 | 0.101 | **0.101** | 0.098 | 0.088 | 0.054 | 0.027 | 0.014 | 0.030 | 0.051 | 0.061 |
| | Max_0% | 0.128 | 0.112 | 0.051 | 0.068 | 0.068 | 0.027 | **0.044** | 0.014 | 0.047 | 0.051 | 0.047 |
| Rényi ($\alpha = \infty$) | Max_10% | 0.142 | 0.112 | 0.068 | 0.091 | 0.074 | 0.027 | 0.020 | 0.017 | 0.034 | 0.054 | 0.034 |
| | Max_100% | 0.024 | **0.122** | 0.081 | **0.098** | **0.095** | 0.037 | 0.034 | 0.014 | 0.024 | 0.064 | 0.041 |

feasibility of this approach on GPT-4, where the top-5 probabilities are available, it can be challenging if only the probability of the target token is provided.

Table 16: Examples in VL-MIA/image non-member data are generated by DALL-E or collected from recent Flickr websites; text non-member data are generated by GPT-4.

| Dataset | Member data | Non-member data |
|---------|-------------|-----------------|
| VL-MIA/DALL-E |  |  |
| |  |  |
| VL-MIA/Flickr |  |  |
| |  |  |
| VL-MIA/Text for MiniGPT-4 | The image shows a bedroom with a wooden headboard and nightstands on either side of the bed. The bed is made with a white comforter and pillows, and there are two lamps | The image shows a bathroom with cream-colored walls. On the left, there is a vanity with a granite countertop and wooden cabinets below. A soap dispenser is placed on the countertop, and |
| | This image shows a blue pickup truck, which appears to be a Volkswagen Beetle, parked in a driveway in front of a house. The hood of the truck is open, exposing the | The image depicts a well-used kitchen with various cooking utensils and food items scattered throughout. On the left, there is a gas stove with a white oven beneath it. Above the stove |
| VL-MIA/Text for LLaVA 1.5 and LLaMA Adapter v2 | To enjoy the last two pieces of cake equally and fairly, I suggest using a knife which, according to the image, is already present on the table. Carefully cut each of the | To enjoy the remaining pieces of this delectable cake fairly, I would recommend dividing the slices equally among those present, ensuring that each person gets an identical portion, or alternatively, one could |
| | The young boy is demonstrating the important habit of maintaining good oral hygiene by brushing his teeth. In the image, he is standing in front of a mirror holding a toothbrush, which | In the image, a young boy is engaging in the imperative habit of brushing his teeth, which is fundamental for maintaining oral hygiene, preventing dental issues like cavities and gum disease, and |

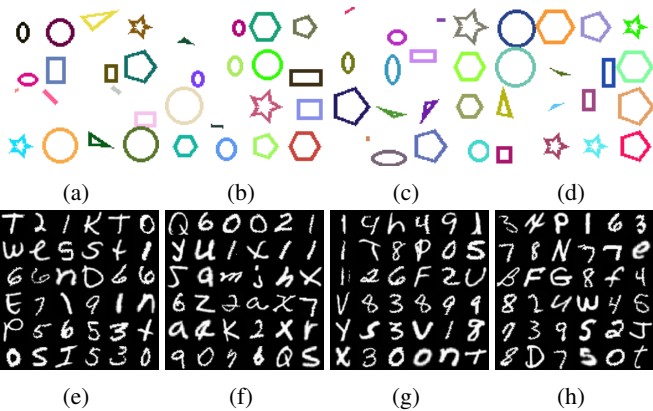

|  (a) | (b) | (c) | (d) |

|  (e) | (f) | (g) | (h) |

Figure 5: Examples of new geometry datasets (top), new password datasets (bottom)..

Table 17: Examples in corrupted versions of VL-MIA/Flickr.

| Severity | Original image | Brightness | Motion_Blur | Snow | JPEG |
| --- | --- | --- | --- | --- | --- |
| Marginal |  |  |  |  |  |
| Moderate |  |  |  |  |  |
| Severe |  |  |  |  |  |

Table 18: Model generated output of corrupted versions of VL-MIA/Flickr. Take severe severity as an example. (`Max_new_tokens` = 64)

| Corruption | Generated output |
| --- | --- |
| None | The image features a large, bright orange truck with a black front end, parked on a dirt road. The truck has a distinctive appearance, as it is equipped with massive tires, giving it a monster truck-like look. The truck is positioned in the middle of |
| Brightness | The image features a large, red four-wheel-drive truck with a black front end, parked on a dirt road. The truck is equipped with massive tires, giving it a distinctive appearance. The truck's front end is adorned with a Ford logo, further |
| Motion_Blur | The image features a large, orange four-wheel-drive truck driving down a dirt road. The truck is the main focus of the scene, occupying a significant portion of the image. The truck appears to be in motion, possibly on a trail or a dirt road. |
| Snow | The image features a large, red pickup truck parked on a dirt road. The truck is equipped with massive tires, giving it a distinctive appearance. The truck is positioned in the middle of the scene, with the front end facing the viewer. |
| JPEG | The image features a large, bright orange truck with a black front end, parked on a dirt road. The truck is equipped with massive tires, giving it a monster-like appearance. The truck is positioned in the center of the scene, with the front end facing the vie |

