# OpenReview forum: "Membership Inference Attacks against Large Vision-Language Models"
_NeurIPS.cc/2024/Conference — NeurIPS 2024 poster_

### Official Review · Reviewer_rMMx · 2024-07-10

**Soundness:** 3
**Presentation:** 3
**Contribution:** 3
**Rating:** 5
**Confidence:** 2

**Summary:**

This paper focuses on the membership inference attack (MIA) of large vision language models. It creates multiple MIA datasets and proposes a new method to identify whether an image or a text belongs to the training dataset.

**Strengths:**

1. The authors have proposed an intriguing research problem, which focuses on the membership attack of large vision language models.
2. The paper proposes a creative method to verify if the input is a member of the training set. It also creates a new benchmark for evaluation.
3. The overall writing is clear and easy to follow.

**Weaknesses:**

The setting for detecting a description sentence is a little confusing for me. Why do you feed the model with a black image rather than others, like those generated by DALL-E? Also, why do you skip the instructions here?  It is less possible to produce the target description for the target VLM just based on a black image and an empty instruction. Why don't skip the input image?

**Questions:**

1. As illustrated in the weakness section, could you please explain the setting of MIA in the description sentence?
2. How to set the threshold value \lambda in the MIA process?

**Limitations:**

The paper has analyzed its limitations in the Appendix, which I believe is reasonable.

---

> ### Author Rebuttal · Authors · 2024-08-06
>
> We thank the reviewer rMMx for the insightful feedback. We address the concerns below.
>
> ---
>
> > **Q1:** [The setting for detecting a description sentence is a little confusing for me. Why do you feed the model with a black image rather than others, like those generated by DALL-E? Also, why do you skip the instructions here? It is less possible to produce the target description for the target VLM just based on a black image and an empty instruction. Why don't skip the input image? Could you please explain the setting of MIA in the description sentence?]
>
> **A1:** Note that our MIA framework is for detecting text or images solely instead of image-description pairs. This setting is more practical. For example, we want to know whether private photo or address is used for commercial training. Naturally, when detecting the text, we assume that the attacker does not have any information about image. Therefore, we can not use images generated from DALL-E.
>
> In line 109, we assume a grey-box setting on the target model, where the attacker can query the model by a custom prompt (including an image and text) and have access to the tokenizer, the output logits, and the generated text. Therefore, we are not allowed to extract the underlying language model from a VLLM. Additionally, if we provide None or empty to VLLM such as llama_adapter v2 in image place, then an error will be raised. Also, we note that in the open source code of llama adapter, when using the text-only data for finetuning, the model is supplied with an all-zero image. For the consistence of our experimental setting that we provide image and text modals as inputs, we choose to provide an all-zero image  (black) to all models.
>
> The reason we adopt an empty instruction is that, provided an all-zero (black) image, common instructions, such as "Describe this image in detail", the VLLMs have the tendency to output response that "This image is black". This will introduce additional bias to our task: the description text MIAs. Therefore, we use the empty instruction.
>
>
> ---
>
>
> > **Q2:** [How to set the threshold value \lambda in the MIA process?]
>
> **A2:** For the evaluation, we use the AUC score and TPR@low FPR score as the metric. The report of these scores as performance measures is standard in prior MIA works, such as [1]. Note that these two scores are threshold-independent. It's the area under ROC curve, which plots the true positive rate against the false positive rate for different threshold settings. So we don't need to select a specific $\lambda$ when we want to compare different MIA methods (classifiers). Note that if the task is to perform MIA via a specific method instead of evaluating different MIA methods, we can perform a hyperparameter search over a small validation set to select $\lambda$. We will add this explanation of the metric in the revised version.
>
>
> ---
>
> If the reviewer rMMx has any remaining concerns, we are happy to clarify further.
>
> ---
>
>
> **Refs**
>
> [1] Shi, Weijia, et al. "Detecting Pretraining Data from Large Language Models." ICLR 2024.

---

> > ### Comment · Reviewer_rMMx · 2024-08-12
> > **Response to authors**
> >
> > Thanks for the detailed response. Although some presentations can still be revised, it is an interesting paper on its threat model design. Therefore, I will maintain my current score.

---

### Official Review · Reviewer_e87W · 2024-07-11

**Soundness:** 3
**Presentation:** 3
**Contribution:** 3
**Rating:** 6
**Confidence:** 3

**Summary:**

This paper presents an interesting idea of detecting training data in large vision-language models (VLLMs) through membership inference attacks (MIAs). The authors introduce the following interesting points :

- An MIA benchmark, specifically designed for VLLMs, called Vision Language MIA (VL-MIA). This benchmark includes tasks for both image and text detection across various VLLM architectures.
- A pipeline for MIAs on VLLMs. This approach does not require both side information as in the prior work. This includes a new detection metric called MaxRényi-K%.
- Extensive evaluation results. The authors demonstrate the effectiveness of their approach to diverse VLLMs.

**Strengths:**

This paper is well-written and proposes an interesting approach that can be applicable to one side of information for MIA for VLLMs, which might not be well-studied in prior works.
The authors also provide a benchmark for evaluating MIA on VLLMs and present extensive results based on the proposed benchmark.

**Weaknesses:**

This paper requires further clarification regarding the proposed methods and evaluation results. Please see the following questions.

**Questions:**

Regarding dataset generation:
If the member data is drawn from LAION_CCS, and non-member data is drawn from the generated dataset, is there any underlying bias caused by the nature of differences between natural and synthetic data (in VL-MIA/DALL-E)? Moreover, why did we choose instruction text for members and generated text for non-members in VT-MIA/Text?

Regarding the evaluation:
Based on Table 2, it seems that different datasets show different behaviors. For example, for VL-MIA/Flickr, Max_100% shows the best performance in most cases, and the performance gap is quite large. On the other hand, we cannot see a similar observation from VL-MIA/DALL-E. I wonder whether the authors have some insight into why it shows a clear difference between datasets.
Moreover,  it seems that we need to vary a lot of parameters to find better performance (e.g., different alpha, and different K). How did the authors choose the hyperparameters from other baselines? Since the authors present a new benchmark, it might not be a good choice to use optimal hyperparameters from other sources.

**Limitations:**

The authors discuss the limitations and broader impacts in the appendix.

---

> ### Author Rebuttal · Authors · 2024-08-06
>
> We thank the reviewer e87W for the insightful feedback. We address the concerns below.
>
> ---
>
> > **Q1:** Is there any underlying bias caused by the nature of differences between natural and synthetic data (in VL-MIA/DALL-E)?
>
> **A1:** This is an interesting question and thanks for pointing it out. It is hard to make the member data and non-member data be in the same distribution. We considered different possible bias when constructing our datasets.
>
> - One bias is that the member data and non-member data might contain different identities so that the separation would be trivial. We call it the identity bias. To resolve the identity bias, we use DALL-E to generate non-member images based on the description of the member images to ensure the member image and non-member image in the same pair contain the same identity, such as a basketball.
>
> - Admittedly, as you pointed out, there might be other unresolved bias, such that the fact that member data are real (not AI generated) images and non-member data are fake (AI generated) images. We call it the reality bias. To resolve the reality bias, we provide "VL-MIA: Flickr", which divide member and non-member images based on release time, following the prior work [1] which uses latest wiki text as non-members.
>
> Despite our effort aforementioned to reduce the bias, we additionally produced two new datasets designed to eliminate bias using i.i.d samples from the same distribution.  We synthesize two new MIA datasets: **the Geometry dataset** and **the Password dataset**. The image in the Geometry dataset consists of a random 4x4 arrangement of geometrical shapes, and the image in the Password dataset consists of a random 6x6 arrangement of characters and digits from (E)MINST. The associated text is its corresponding content (e.g., characters, colors, shapes). We select half of the datasets, that can be considered as the member set, to finetune VLLM while the remaining part is the non-member set. This approach ensures that members and non-members are **i.i.d sampled** thus eliminating potential bias, and can be used in **any VLLM** with diversity and generalization just by simple finetuning. We provide some examples of this dataset in [Figure 2 of the PDF](https://openreview.net/forum?id=nv2Qt5cj1a&noteId=4ygbDFCzeE).
>
> We conduct the image MIA by finetuning LLaMA Adapter v2.1 following the finetune instruction provided by the authors. The results are presented in [Table 4 of the PDF](https://openreview.net/forum?id=nv2Qt5cj1a&noteId=4ygbDFCzeE). We observe that our Modfied Rényi and Max Rényi still outperform other previous baselines.
>
>
> ---
>
> > **Q2:** [why did we choose instruction text for members and generated text for non-members in VT-MIA/Text?]
>
> **A2:** Sorry for the confusion, we'll clarify in the final release. Recall that in the instruction tuning stage of a vllm, every data entry contains an image, a question, and the corresponding answer to the image. We use the answer text as member data and GPT-4 generated answers under the same question and same image as non-member data. Specifically, for LLaVA v1.5 and LLaMA_adapter v2, we use the answers in LLaVA v1.5's instruction tuning as member data. (The word "instruction" in paper main body Table 1 is a typo and will be corrected as "instruction-tuning text".) For MiniGPT-4, we use answers in MiniGPT-4's instruction tuning as member data.  We will revise the confusing content in Table 1 in the final release.
>
> ---
>
> > **Q3:** Insight on the parameters $\alpha$ and $K$. How to select these parameters?
>
> **A3:** In short, $\alpha$ controls the aggregation from the next-token distribution at some token to a single entropy score, and $K$ controls the aggregation from a sequence of entropy scores to the final score for this sequence.
>
> - The parameter $\alpha$ controls how the entropy will represent the next token probability distribution at the current token. For example, $\alpha=0$ treats all non-zero next-token probabilities equally and $\alpha=\infty$ only involves the largest next-token probability.
> In paper main body Table 2 we find that $\alpha = 0.5$ is the best choice in image MIA, while in paper main body Table 3, $\alpha = \infty$ is the best choice in text MIA.
>
> - The parameter $K$ determines the percentage of tokens in a sequence with the highest entropy scores that are selected to compute the final score. In the experiment section of the paper, we find that different sequences have different representative parts, and therefore different $K$ may be applied.
>
> In all, the optimal $K$ and $\alpha$ are largely determined by different data modalities and distributions. We propose this family of criteria to accommodate different possible data distributions. Therefore, similar to prior work [1], we suggest using a validation set and performing a small sweep over different $K$ and $\alpha$ to select the optimal parameters.
>
> ---
>
> If the reviewer e87W has any remaining concerns, we are happy to clarify further.
>
> ---
>
> **Refs**
>
> [1] "Detecting Pretraining Data from Large Language Models." ICLR 2024.

---

> > ### Comment · Reviewer_e87W · 2024-08-11
> > **response to authors**
> >
> > Thank the authors for your efforts in responding to questions and conducting additional experiments. I still have some questions as follows:
> >
> > In the Text MIA experiments along with DALL-E, the authors compare generated data and real-world data, which might already contain underlying biases that could affect performance. I understand that the authors also used Flickr by splitting the data between the target model's release date, and it has even shown better performance with the authors' proposed method (e.g., Flickr has consistently outperformed results from Renyi) than others. Also, as shown in the additional results from the PDF, the performance difference on synthetic data seems marginal.
> >
> > In this case, which dataset should we rely on more if the two datasets show different trends?
> >
> > For example, if we can find one real-world text pair (before and after the target models' release date), and it also shows better performance with the proposed method, as presented in image experiments, why is there a difference? Which one should we trust more if the authors want to present a benchmark?

---

> > > ### Author Response · Authors · 2024-08-12
> > > **response to reviewer e87W**
> > >
> > > We appreciate the insightful feedback from reviewer e87W. Note that our results across all datasets exhibit a consistent trend. Specifically, in Table 4 of the rebuttal PDF, our proposed Rényi metric demonstrates a considerable improvement in AUC for the newly constructed synthetic dataset: from 0.65 to 0.69 with the description slice, and from 0.55 to 0.65 with the instruction+description slice, compared to the previous baselines. From Table 2 of our paper, we also notice that "Rényi ($\alpha = 0.5$)" generally outperforms previous baselines in image MIA for both Flickr and Dalle-E datasets.
> > >
> > > Regarding the choice of datasets, we believe **both** the synthetic dataset and the Flickr dataset **should be considered**. Previous literature in machine learning [1,2,3] suggest that evaluating across multiple datasets, such as MNIST, CIFAR-10, ImageNet, and synthetic (Gaussian) data, can demonstrate its effectiveness in various scenarios. In our benchmark, the synthetic and real-world dataset have their own benefit. The advantage of our synthetic geometry and password dataset is that it adheres to the i.i.d. assumption of member and non-member data, and the membership can be fully determined. Meanwhile, the Flickr dataset aligns more closely with the real-world data distributions. We verify our method's performance in both synthetic and real-world datasets.
> > >
> > > ---
> > > Reference:
> > >
> > > [1] "Membership inference attacks against machine learning models." 2017.
> > >
> > > [2] "Descending through a crowded valley-benchmarking deep learning optimizers." 2021.
> > >
> > > [3] "ViLLA: Fine-Grained Vision-Language Representation Learning from Real-World Data." 2023.

---

### Official Review · Reviewer_FZ3m · 2024-07-11

**Soundness:** 3
**Presentation:** 2
**Contribution:** 3
**Rating:** 6
**Confidence:** 3

**Summary:**

The paper introduced a benchmark for membership inference attack on VLMs, proposed a pipeline for token-level image detection, and proposed a target-free metric for image MIA detections. The pipeline relies on the fixed sequence of the VLM output to obtain the output image, instruction, and description segments of logits and use them for evaluation metrics.

**Strengths:**

1. Novelty: There is no existing MIA benchmark datasets for VLM. The authors also proposed a new metrics for detecting MIA in single modality, especially in images.
2. The paper conducts extensive experiments with their proposed methods.

**Weaknesses:**

1. The explanation of some concepts are not clear. For example, when proposing target-free MIA metrics, the author mentioned that it's because we only have access to the image embeddings but not image tokens. How are these two terms defined here? Why do we not have access to the image tokens? Additionally, image tokens are defined again in line 95. If this is not available, why does the paper need to define the concept again? I think the authors should clarify these assumptions properly in the main text.

**Questions:**

1. For the cross-modal pipeline for detecting images, how do you determine which logits are for image, instructions, or description text, during the attack phase?
2. The proposed metrics can have different K and $\alpha$ values. Are there any ablation studies on when should what K and $\alpha$ values being used?

**Limitations:**

The authors clarified the limitations in the paper.

---

> ### Author Rebuttal · Authors · 2024-08-06
>
> We thank the reviewer FZ3m for the insightful feedback. We address the concerns below.
>
> ---
>
> > **Q1:** Clarification on image embeddings but not image tokens. Why do we not have access to the image tokens?
>
> **A1:** As in the illustrative [Figure 1 of the PDF](https://openreview.net/forum?id=nv2Qt5cj1a&noteId=4ygbDFCzeE), the VLLMs consist of a vision encoder, a text tokenizer, and a language model. The output of the vision encoder (image embedding) has the same embedding dimension $d$ as the text token embedding.
>
> When feeding the VLLMs with an image and the instruction, a vision encoder transforms this image to $L_1$ hidden embeddings of dimension $d$, denoted by $E_{\rm image} \in \mathbb{R}^{d \times L_1}$. The text tokenizer first tokenizes the instruction into $L_2$ tokens and then looks up the embedding matrix to get its $d$-dimensional  embedding $E_{\rm ins} \in \mathbb{R}^{d \times L_2}$. The image embedding and the instruction embedding are concatenated as  $E_{\rm img-ins} = (E_{\rm image}, E_{\rm ins}) \in \mathbb{R}^{d \times (L_1+L_2)}$, which are then fed into a language model to perform next token prediction. The cross-entropy loss (CE loss) is calculated based on the predicted token id and the ground truth token id on the text tokens.
>
> We can see that in this process, image embeddings are directly obtained from the vision encoder and there are no image tokens. There are no causal relations between consecutive image embeddings as well. Therefore, as we stated in Section 1, target-based MIA that requires token ids cannot be directly applied.
>
> In the final version, we will add the detailed description as well as the illustrative figure in the appendix.
>
> ---
>
> > **Q2:** For detecting images, how do you determine which logits are for image, instructions, or description text, during the attack phase?
>
> **A2:** Similarly to the figure in Q1, there is a one-to-one correspondence between the logit and input. For example, given image embedding with token length $L_1$,  instructions with length $L_2$, and description text with length $L_3$, the language model will output logits of the shape $(L_1+L_2+L_3)\times |\mathcal{V}|$, where $\mathcal{V}$ is the vocabulary set, we can access the logits of the image by the slice $[0:L_1]$, the logits of instruction by the slice $[L_1:L_1+L_2]$, and the logits of description by the slice $[L_1+L_2: L_1+L_2+L_3]$.
>
>
> ---
>
>
> > **Q3:** How to choose $K$ and $\alpha$ for the MaxRényi-K% metric?
>
> **A3:** In short, $\alpha$ controls the aggregation from the next-token distribution at some token to a single entropy score, and $K$ controls the aggregation from a sequence of entropy scores to the final score for this sequence.
>
> - The parameter $\alpha$ controls how the entropy will represent the next token probability distribution at the current token. For example, $\alpha=0$ treats all non-zero next-token probabilities equally and $\alpha=\infty$ only involves the largest next-token probability.
> In paper main body Table 2 we find that $\alpha = 0.5$ is the best choice in image MIA, while in paper main body Table 3, $\alpha = \infty$ is the best choice in text MIA.
>
> - The parameter $K$ determines the percentage of tokens in a sequence with the highest entropy scores that are selected to compute the final score. In the experiment section of the paper, we find that different sequences have different representative parts, and therefore different $K$ may be applied.
>
> In all, the optimal $K$ and $\alpha$ are largely determined by different data modalities and distributions. We propose this family of criteria to accommodate different possible data distributions. Therefore, similar to prior work [1], we suggest using a validation set and performing a small sweep over different $K$ and $\alpha$ to select the optimal parameters.
>
> ---
>
> If the reviewer FZ3m has any remaining concerns, we are happy to clarify further.
>
> ---
>
> **Refs**
>
> [1] "Detecting Pretraining Data from Large Language Models." ICLR 2024.

---

### Official Review · Reviewer_M6Nm · 2024-07-12

**Soundness:** 3
**Presentation:** 3
**Contribution:** 2
**Rating:** 5
**Confidence:** 3

**Summary:**

The rise of large vision-language models (VLLMs) has significantly advanced multi-modal tasks but also brought forth concerns about data security and privacy. This paper introduces a novel membership inference attack (MIA) benchmark specifically designed for VLLMs to detect training data, addressing the lack of standardized datasets and methodologies in this domain. The authors propose a new MIA pipeline for token-level image detection and introduce the MaxRényi-K% metric for improved detection. The key contributions include the development of the first VLLM-specific MIA benchmark, a cross-modal MIA pipeline for individual image or description detection, and the new MaxRényi-K% and ModRényi metrics, which show effectiveness in experiments.

**Strengths:**

- The paper introduces the first benchmark specifically tailored for VLLMs in the context of MIAs.
- The MaxRényi-K% and ModRényi metrics demonstrate significant effectiveness in detecting training data.

**Weaknesses:**

- **Small Evaluation Dataset**: The MIA evaluation dataset consists of around 600 images for each evaluation, which is relatively small. This limited dataset size can lead to less statistically robust results and may not fully capture the variability and challenges present in real-world scenarios. It would strengthen the paper to include evaluations on larger and more diverse datasets.

- **Lack of Standard Metrics**: The paper does not consider the _TPR at low FPR_ (True Positive Rate at low False Positive Rate) metric, which is a standard for evaluating worst-case membership privacy. Including this metric would provide a more comprehensive assessment of the model's privacy risks, especially in high-stakes applications where false positives must be minimized.

- **Generalizability**: While the proposed metrics and methods show effectiveness, it's important to discuss their generalizability to other types of vision-language models and datasets. Providing insights or experiments on different architectures or domains could enhance the applicability of the findings.

**Questions:**

- How to choose $K$ for the MaxRényi-K% metric?

**Limitations:**

The size and the diversity of the benchmark is limited.

---

> ### Author Rebuttal · Authors · 2024-08-06
>
> We thank the reviewer M6Nm for the insightful feedback. We address the concerns below.
>
> ---
>
> > **Q1:** Small Evaluation Dataset.
>
>
> **A1:**  We extend both VL-MIA/Flickr and VL-MIA/Text to 2000 samples. The results in the extended datasets can be found in [Table 2 and Table 3 of the PDF](https://openreview.net/forum?id=nv2Qt5cj1a&noteId=4ygbDFCzeE), where we can observe the same trend as our original results. We also introduce two new datasets as shown in A3 below. We will release these complete datasets in the final version.
>
> > **Q2:** Does not consider the TPR at low FPR.
>
> **A2:**  Please see TPR at 5% FPR result in [Table 1 of the PDF](https://openreview.net/forum?id=nv2Qt5cj1a&noteId=4ygbDFCzeE). We will add the results in our final version. These results are aligned with AUC results in Table 1 of the paper, and $\alpha=0.5$ is the optimal choice for image MIA.
>
> > **Q3:** Generalizability to other types of vision-language models and datasets.
>
> **A3:** We thank the reviewer for pointing out this interesting question. To make our benchmark more comprehensive, we synthesize two new MIA datasets, see [general response](https://openreview.net/forum?id=nv2Qt5cj1a&noteId=4ygbDFCzeE) for details.
>
> ---
>
> > **Q4:** How to choose $K$ for the MaxRényi-K% metric?
>
> **A4:** In short, $K$ controls the aggregation from a sequence of entropy scores to the final score for this sequence.
>
> The parameter $K$ determines the percentage of tokens in a sequence with the highest entropy scores that are selected to compute the final score. In the experiment section of the paper, we find that different sequences have different representative parts, and therefore different $K$ may be applied.
>
> In all, the optimal $K$ is largely determined by different data modalities and distributions. We propose this family of criteria to accommodate different possible data distributions. Therefore, similar to prior work [1], we suggest using a validation set and performing a small sweep over different $K$ and $\alpha$ to select the optimal parameters.
>
> ---
> If the reviewer M6Nm has any remaining concerns, we are happy to clarify further.
>
> ---
>
> **Refs**
>
> [1] "Detecting Pretraining Data from Large Language Models." ICLR 2024.

---

> > ### Comment · Reviewer_M6Nm · 2024-08-10
> > **Thank you for the response**
> >
> > Thanks for the response and the updated results.
> >
> > Using a validation dataset to tune hyperparameters is a reasonable approach. However, this method assumes that the attacker has collected some member and non-member instances in advance. I’m curious about how practical this assumption is in real-world scenarios. I would appreciate it if the authors could provide further clarification on this point.

---

> ### Author Response · Authors · 2024-08-10
> **Clarification on the assumption**
>
> We thank reviewer M6Nm for the meaningful feedback. Regarding the concern that our method assumes that the attacker has collected some member and non-member data in advance, we explain that this assumption is practical in real-world scenarios. For open-source models trained on open-source data, we can confidently obtain both member and non-member data, as demonstrated in this paper's Section 4. For closed-source models, such as GPT4, we speculate that commonly used datasets, such as MS COCO, and a wide collection of copyrighted materials [1], are used as training data by these closed-source models.
>
> Note that all of the baseline methods (e.g., PPL, min-k) also require a validation set to estimate a threshold ($\lambda$ in Equation 1) to determine membership. The above choice of member and non-member dataset can be universally used for all MIA methods.
>
> We would like to add these clarifications after line 305 with the revised version to enhance understanding on member and non-member data selection.
>
> Refs:
>
> [1] "Speak, memory: An archaeology of books known to chatgpt/gpt-4." arXiv 2023.
>
> ---
> Remark: We have further **extended our benchmark size to 10k** and observed similar results, which will be released in the final version. In addition, our heuristic findings in the paper suggest that for text MIA, one can select a smaller k, e.g.,0 or 10; while for image MIA, one can select k=100.

---

> > ### Comment · Reviewer_M6Nm · 2024-08-11
> > **Thank you**
> >
> > Thank you for the reply. It appears that the threshold-based MIAs rely on somewhat strong assumptions, particularly the need for a validation dataset, which may limit their broader applicability. However, I still believe this paper offers valuable new insights into the privacy issue of vision-language models.
> >
> > I will maintain my current score.

---

### Author Rebuttal · Authors · 2024-08-06

Dear reviewers,

We appreciate your insightful comments. The attached PDF contains the necessary figures and tables corresponding to each individual response below.

During the rebuttal period, we expand our benchmark by incorporating new diverse datasets, as motivated by reviewers **e87W** and **M6Nm**. Specifically, in order to make our benchmark more comprehensive, we synthesize two new MIA datasets: **the Geometry dataset** and **the Password dataset**. The image in the Geometry dataset consists of a random 4x4 arrangement of geometrical shapes, and the image in the Password dataset consists of a random 6x6 arrangement of characters and digits from EMINST [1] and MNIST. The associated text is its corresponding content (e.g., characters, colors, shapes). We select half of the datasets, that can be considered as the member set, to finetune VLLM while the remaining part is the non-member set. This approach ensures that members and non-members are **i.i.d sampled** thus eliminating potential bias, and can be used in **any VLLM** with diversity and generalization just by simple finetuning. We provide some examples of this dataset in Figure 2 of the PDF.

We conduct the image MIA by finetuning LLaMA Adapter v2.1 following the finetune instruction provided by the authors. The results are presented in Table 4 of the PDF. We observe that our Modfied Rényi and Max Rényi still outperform other previous baselines.

---
Refs

[1] "EMNIST: Extending MNIST to handwritten letters." IJCNN, 2017.

---

### Decision · Program_Chairs · 2024-09-25

**Decision:**

Accept (poster)

**Comment:**

The paper introduces the first benchmark specifically for MLA on LVLMs and proposes novel metrics and methods for detecting training data usage. Reviewers found the research problem significant and the approach innovative, appreciating the extensive experiments and novel benchmarks. However, concerns were raised about dataset size, the clarity of some methodological choices, and the practical applicability of the results. The authors' rebuttal provided additional datasets, expanded evaluations, and further clarifications, addressing most concerns satisfactorily. The recommendation is to accept.